# Language Models are Few-Shot Learners

**Tom B. Brown**∗  **Benjamin Mann**∗  **Nick Ryder**∗  **Melanie Subbiah**∗

**Jared Kaplan**†  **Prafulla Dhariwal**  **Arvind Neelakantan**  **Pranav Shyam**

**Girish Sastry**  **Amanda Askell**  **Sandhini Agarwal**  **Ariel Herbert-Voss**

**Gretchen Krueger**  **Tom Henighan**  **Rewon Child**  **Aditya Ramesh**

**Daniel M. Ziegler**  **Jeffrey Wu**  **Clemens Winter**

**Christopher Hesse**  **Mark Chen**  **Eric Sigler**  **Mateusz Litwin**  **Scott Gray**

**Benjamin Chess**  **Jack Clark**  **Christopher Berner**

**Sam McCandlish**  **Alec Radford**  **Ilya Sutskever**  **Dario Amodei**

## Abstract

We demonstrate that scaling up language models greatly improves task-agnostic, few-shot performance, sometimes even becoming competitive with prior state-of-the-art fine-tuning approaches. Specifically, we train GPT-3, an autoregressive language model with 175 billion parameters, 10x more than any previous non-sparse language model, and test its performance in the few-shot setting. For all tasks, GPT-3 is applied without any gradient updates or fine-tuning, with tasks and few-shot demonstrations specified purely via text interaction with the model. GPT-3 achieves strong performance on many NLP datasets, including translation, question-answering, and cloze tasks. We also identify some datasets where GPT-3's few-shot learning still struggles, as well as some datasets where GPT-3 faces methodological issues related to training on large web corpora.

## 1  Introduction

NLP has shifted from learning task-specific representations and designing task-specific architectures to using task-agnostic pre-training and task-agnostic architectures. This shift has led to substantial progress on many challenging NLP tasks such as reading comprehension, question answering, textual entailment, among others. Even though the architecture and initial representations are now task-agnostic, a final task-specific step remains: fine-tuning on a large dataset of examples to adapt a task agnostic model to perform a desired task.

Recent work [RWC+19] suggested this final step may not be necessary. [RWC+19] demonstrated that a single pretrained language model can be zero-shot transferred to perform standard NLP tasks

---

∗Equal contribution

†Johns Hopkins University, OpenAI

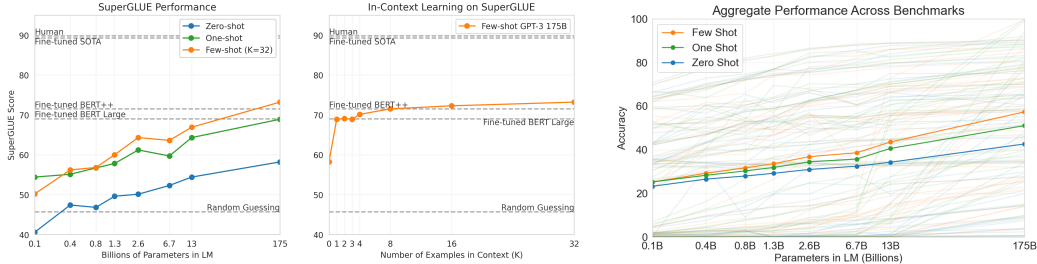

**Figure 1.1: Performance on SuperGLUE increases with model size.** A value of $K = 32$ means that our model was shown 32 examples per task, for 256 examples total divided across the 8 tasks in SuperGLUE. We report GPT-3 values on the dev set, so our numbers are not directly comparable to the dotted reference lines (our test set results are in the appendix). The BERT-Large reference model was fine-tuned on the SuperGLUE training set (125K examples), whereas BERT++ was first fine-tuned on MultiNLI (392K examples) and SWAG (113K examples) before further fine-tuning on the SuperGLUE training set (for a total of 630K fine-tuning examples).
**Performance on SuperGLUE increases with number of examples in context.** We find the difference in performance between the BERT-Large and BERT++ to be roughly equivalent to the difference between GPT-3 with one example per context versus eight examples per context.
**Aggregate performance for all 42 accuracy-denominated benchmarks.** While zero-shot performance improves steadily with model size, few-shot performance increases more rapidly, demonstrating that larger models are more proficient at in-context learning.

without the need for finetuning on a dataset of training examples. While this work was a promising proof of concept, the best case performance only matched some supervised baselines on a single dataset. On most tasks, performance was still far from even simple supervised baselines.

However [RWC+19] also showed a potential way forward. The work observed relatively consistent log-linear trends in performance on both transfer tasks and language modeling loss across one an order of magnitude of scaling. [KMH+20] then conducted a much more rigorous study of the scaling behavior of log loss and confirmed smooth scaling trends. In this work, we empirically test whether scaling continues to improve performance by extrapolating the previously identified phenomena another two orders of magnitude. We train a 175 billion parameter autoregressive language model, which we call GPT-3, and measure its transfer learning abilities.

As part of this investigation, we also clarify and systematize the approach introduced in [RWC+19]. While [RWC+19] describe their work as "zero-shot task transfer" they sometimes provide examples of the relevant task in the context. Due to the use of what are effectively training examples, these cases are better described as "one-shot" or "few-shot" transfer. We study these one-shot and few-shot settings in detail comparing them with the zero-shot setting which only uses a natural language description or invocation of the task to be performed. Our findings are summarized in Figure 1.1. We observe that one- and few-shot performance is often much higher than true zero-shot performance leading us to suggest that language models can also be understood as meta-learners where slow outer-loop gradient descent based learning is combined with fast "in-context" learning implemented within the context activations of the model.

Broadly, on NLP tasks GPT-3 achieves promising results in the zero- and one-shot settings, and in the few-shot setting is sometimes competitive with or even occasionally surpasses state-of-the-art (despite state-of-the-art being held by fine-tuned models). For example, GPT-3 achieves 81.5 F1 on CoQA in the zero-shot setting, 84.0 F1 on CoQA in the one-shot setting, and 85.0 F1 in the few-shot setting. Similarly, GPT-3 achieves 64.3% accuracy on TriviaQA in the zero-shot setting, 68.0% in the one-shot setting, and 71.2% in the few-shot setting, the last of which is state-of-the-art relative to fine-tuned models operating in the same closed-book setting.

We additionally train a series of smaller models (ranging from 125 million parameters to 13 billion parameters) in order to compare their performance to GPT-3 in the zero-, one- and few-shot settings. In general, we find relatively smooth scaling for most tasks with model capacity in all three settings; one notable pattern is that the gap between zero-, one-, and few-shot performance often grows with model capacity, perhaps suggesting that larger models are more proficient meta-learners.

## 2   Approach

Our basic pre-training approach, including model, data, and training, is similar to the process described in [RWC⁺19], with relatively straightforward scaling up of the model size, dataset size and diversity, and length of training. Our use of in-context learning is also similar to [RWC⁺19], but in this work we systematically explore different settings for learning within the context:

- **Fine-Tuning (FT)** - updates the weights of a pre-trained model by training on thousands of supervised labels specific to the desired task. The main advantage of fine-tuning is strong performance on many benchmarks. The main disadvantages are the need for a new large dataset for every task, the potential for poor generalization out-of-distribution [MPL19], and the potential to exploit spurious features of the training data [GSL⁺18, NK19]. We focus on task-agnostic performance, leaving fine-tuning for future work.

- **Few-Shot (FS)** - the model is given a few demonstrations of the task at inference time as conditioning [RWC⁺19], but no weights are updated. An example typically has a context and a desired completion (for example an English sentence and the French translation), and few-shot works by giving $K$ examples of context and completion, and then one final example of context, with the model expected to provide the completion (see appendix for more details). We typically set $K$ in the range of 10 to 100, as this is how many examples can fit in the model's context window ($n_{\text{ctx}} = 2048$). The main advantage of few-shot is a major reduction in the need for task-specific data. The main disadvantage is that results from this method have so far been much worse than state-of-the-art fine-tuned models. Also, a small amount of task specific data is still required. As indicated by the name, few-shot learning as described here for language models is related to few-shot learning as used in other contexts in ML [HYC01, VBL⁺16] – both involve learning based on a broad distribution of tasks and then rapidly adapting to a new task.

- **One-Shot (1S)** - similar to few-shot but with $K = 1$.

- **Zero-Shot (0S)** - similar to few-shot but with a natural language description of the task instead of any examples.

The appendix includes a demonstration of the four methods using the example of translating English to French. While the few-shot results we present in this paper achieve the highest performance, one-shot, or even sometimes zero-shot, seem like the fairest comparisons to human performance, and are important targets for future work.

### 2.1   Model and Architectures

We use the same model and architecture as GPT-2 [RWC⁺19], including the modified initialization, pre-normalization, and reversible tokenization described therein, with the exception that we use alternating dense and locally banded sparse attention patterns in the layers of the transformer, similar to the Sparse Transformer [CGRS19]. To study the dependence of ML performance on model size, we train 8 different sizes of model, from 125 million parameters to 175 billion parameters, with the last being the model we call GPT-3. This range of model sizes allows us to test the scaling laws introduced in [KMH⁺20].

More details on the sizes and architectures of our models can be found in the appendix. We partition each model across GPUs along both the depth and width dimension in order to minimize data-transfer between nodes.

### 2.2   Training Dataset

To create our training data, we (1) downloaded and filtered a version of CommonCrawl[1] [RSR⁺19] based on similarity to a range of high-quality reference corpora, (2) performed fuzzy deduplication at the document level, within and across datasets, to prevent redundancy and preserve the integrity of our held-out validation set as an accurate measure of overfitting, and (3) added known high-quality reference corpora to the training mix to augment CommonCrawl and increase its diversity. These reference corpora include an expanded version of the WebText dataset [RWC⁺19], collected by

| Setting | LAMBADA (acc) | LAMBADA (ppl) | StoryCloze (acc) | HellaSwag (acc) |
|---|---|---|---|---|
| SOTA | 68.0[a] | 8.63[b] | **91.8[c]** | **85.6[d]** |
| GPT-3 Zero-Shot | **76.2** | **3.00** | 83.2 | 78.9 |
| GPT-3 One-Shot | **72.5** | **3.35** | 84.7 | 78.1 |
| GPT-3 Few-Shot | **86.4** | **1.92** | 87.7 | 79.3 |

**Table 3.1: Performance on cloze and completion tasks.** GPT-3 significantly improves SOTA on LAMBADA while achieving respectable performance on two difficult completion prediction datasets. [a][Tur20] [b][RWC+19] [c][LDL19] [d][LCH+20]

scraping links over a longer period of time, and first described in [KMH+20], two internet-based books corpora (Books1 and Books2) and English-language Wikipedia (details in the appendix).

## 2.3 Training Process

As found in [KMH+20, MKAT18], larger models can typically use a larger batch size, but require a smaller learning rate. We measure the gradient noise scale during training and use it to guide our choice of batch size [MKAT18]. Table A.1 shows the parameter settings we used. To train the larger models without running out of memory, we use a mixture of model parallelism within each matrix multiply and model parallelism across the layers of the network. All models were trained on V100 GPU's on part of a high-bandwidth cluster. Details of the training process and hyperparameter settings are described in the appendix.

## 2.4 Evaluation

For few-shot learning, we evaluate each example in the evaluation set by randomly drawing $K$ examples from that task's training set as conditioning, delimited by 1 or 2 newlines depending on the task. For LAMBADA and Storycloze there is no supervised training set available so we draw conditioning examples from the development set and evaluate on the test set.

For some tasks we use a natural language prompt in addition to (or for $K = 0$, instead of) demonstrations. Similar to [RSR+19] we also sometimes change the formatting of answers. See the appendix for per-task examples.

On tasks with free-form completion, we use beam search with the same parameters as [RSR+19]: a beam width of 4 and a length penalty of $\alpha = 0.6$.

Final results are reported on the test set when publicly available, for each model size and learning setting (zero-, one-, and few-shot). When the test set is private, our model is often too large to fit on the test server, so we report results on the development set.

# 3 Results

## 3.1 Language Modeling, Cloze, and Completion Tasks

We test GPT-3's performance on the traditional task of language modeling as well as related tasks. We calculate zero-shot perplexity on the Penn Tree Bank (PTB) [MKM+94] dataset measured in [RWC+19]. We omit the 4 Wikipedia-related tasks and the one-billion word benchmark due to a high fraction of these datasets being contained in our training set. Our largest model sets a new SOTA on PTB by a substantial margin of 15 points.

The LAMBADA dataset [PKL+16] requires the model to predict the last word of a paragraph. Although [BHT+20] suggested scaling language models is yielding diminishing returns on this benchmark, we find that zero-shot GPT-3 achieves a substantive gain of 8% over the previous state-of-the-art. For the few-shot setting, we use a fill-in-the-blank format to encourage the language model to only generate one word (*Alice was friends with Bob. Alice went to visit her friend, ______. → Bob*). With this format, GPT-3 achieves an increase of over 18% from the previous state-of-the-art, and

| Setting | NaturalQS | WebQS | TriviaQA |
|---|---|---|---|
| RAG (Fine-tuned, Open-Domain) [LPP+20] | **44.5** | **45.5** | 68.0 |
| T5-11B+SSM (Fine-tuned, Closed-Book) [RRS20] | 36.6 | 44.7 | 60.5 |
| T5-11B (Fine-tuned, Closed-Book) | 34.5 | 37.4 | 50.1 |
| GPT-3 Zero-Shot | 14.6 | 14.4 | 64.3 |
| GPT-3 One-Shot | 23.0 | 25.3 | **68.0** |
| GPT-3 Few-Shot | 29.9 | 41.5 | **71.2** |

**Table 3.2: Results on three Open-Domain QA tasks.** GPT-3 is shown in the few-, one-, and zero-shot settings, as compared to prior SOTA results for closed book and open domain settings. TriviaQA few-shot result is evaluated on the wiki split test server.

| Setting | ARC (Easy) | ARC (Challenge) | CoQA | DROP |
|---|---|---|---|---|
| Fine-tuned SOTA | **92.0**[a] | **78.5**[b] | **90.7**[c] | **89.1**[d] |
| GPT-3 Zero-Shot | 68.8 | 51.4 | 81.5 | 23.6 |
| GPT-3 One-Shot | 71.2 | 53.2 | 84.0 | 34.3 |
| GPT-3 Few-Shot | 70.1 | 51.5 | 85.0 | 36.5 |

**Table 3.3:** GPT-3 results on a selection of QA / RC tasks. CoQA and DROP are F1 while ARC reports accuracy. See the appendix for additional experiments. [a][KKS+20] [b][KKS+20] [c][JZC+19] [d][JN20]

performance improves smoothly with model size. However, the fill-in-blank method is not effective one-shot, where it always performs worse than the zero-shot setting, perhaps because all models require several examples to recognize the pattern. An analysis of test set contamination identified that a significant minority of the LAMBADA dataset appears to be present in our training data – however analysis performed in Section 4 suggests negligible impact on performance.

The HellaSwag dataset [ZHB+19] involves picking the best ending to a story or set of instructions. The examples were adversarially mined to be difficult for language models while remaining easy for humans. GPT-3 outperforms a fine-tuned 1.5B parameter language model [ZHR+19] but is still a fair amount lower than the overall SOTA achieved by the fine-tuned multi-task model ALUM.

The StoryCloze 2016 dataset [MCH+16] involves selecting the correct ending sentence for five-sentence long stories. Here GPT-3 improves over previous zero-shot results by roughly 10% but is overall still 4.1% lower than the fine-tuned SOTA using a BERT based model [LDL19].

## 3.2 Question Answering

In this section we measure GPT-3's ability to handle a variety of question answering tasks. First, we look at datasets involving answering questions about broad factual knowledge. We evaluate in the "closed-book" setting (meaning no conditioning information/articles) as suggested by [RRS20]. On TriviaQA [JCWZ17], GPT-3 zero-shot already outperforms the fine-tuned T5-11B by 14.2%, and also outperforms a version with Q&A tailored span prediction during pre-training by 3.8%. The one-shot result improves by 3.7% and matches the SOTA for an open-domain QA system which not only fine-tunes but also makes use of a learned retrieval mechanism over a 15.3B parameter dense vector index of 21M documents [LPP+20]. GPT-3's few-shot result further improves performance another 3.2% beyond this. On Natural Questions (NQs) [KPR+19], GPT-3 underperforms a fine-tuned T5 11B+SSM. The questions in NQs tend towards fine-grained Wikipedia knowledge which could be testing the limits of GPT-3's capacity and broad pretraining distribution.

ARC [CCE+18] is a common sense reasoning dataset of multiple-choice questions collected from 3rd to 9th grade science exams. On the "Challenge" version of the dataset, which has been filtered to questions which simple statistical or information retrieval methods are unable to correctly answer, GPT-3 approaches the performance of a fine-tuned RoBERTa baseline [KKS+20]. On the "Easy"

| Setting | En→Fr | Fr→En | En→De | De→En | En→Ro | Ro→En |
|---|---|---|---|---|---|---|
| SOTA (Supervised) | **45.6**[a] | 35.0 [b] | **41.2**[c] | 40.2[d] | **38.5**[e] | **39.9**[e] |
| XLM [LC19] | 33.4 | 33.3 | 26.4 | 34.3 | 33.3 | 31.8 |
| MASS [STQ+19] | _37.5_ | 34.9 | 28.3 | 35.2 | _35.2_ | 33.1 |
| mBART [LGG+20] | - | - | _29.8_ | 34.0 | 35.0 | 30.5 |
| GPT-3 Zero-Shot | 25.2 | 21.2 | 24.6 | 27.2 | 14.1 | 19.9 |
| GPT-3 One-Shot | 28.3 | 33.7 | 26.2 | 30.4 | 20.6 | 38.6 |
| GPT-3 Few-Shot | 32.6 | _39.2_ | 29.7 | _40.6_ | 21.0 | _39.5_ |

**Table 3.4: Few-shot GPT-3 outperforms previous unsupervised NMT work by 5 BLEU when translating into English reflecting its strength as an English LM.** We report BLEU scores on the WMT'14 Fr↔En, WMT'16 De↔En, and WMT'16 Ro↔En datasets as measured by `multi-bleu.perl` with XLM's tokenization in order to compare most closely with prior unsupervised NMT work. SacreBLEU[f] [Pos18] results reported in the appendix. Underline indicates an unsupervised or few-shot SOTA, bold indicates supervised SOTA with relative confidence. [a][EOAG18] [b][DHKH14] [c][WXH+18] [d][oR16] [e][LGG+20] [f][SacreBLEU signature: BLEU+case.mixed+numrefs.1+smooth.exp+tok.intl+version.1.2.20]

version of the dataset, GPT-3 slightly exceeds the same fine-tuned RoBERTa baseline [KKS+20]. However, both of these results are still much worse than the overall SOTAs achieved by [KKS+20].

Finally, we evaluate GPT-3 on two reading comprehension datasets. Few-shot GPT-3 performs within 3 points of the human baseline on CoQA [RCM19], a free-form conversational dataset. On DROP [DWD+19], a dataset testing discrete reasoning and numeracy, few-shot GPT-3 outperforms the fine-tuned BERT baseline from the original paper but is still well below both human performance and state-of-the-art approaches which augment neural networks with symbolic systems [RLL+19].

## 3.3 Translation

In collecting training data for GPT-3, we used the unfiltered distribution of languages reflected in internet text datasets (primarily Common Crawl). As a result, although GPT-3's training data primarily consists of English (93% by word count), it also includes 7% non-English content (full list at GPT-3 GitHub). Existing unsupervised machine translation approaches often combine pretraining on a pair of monolingual datasets with back-translation [SHB15] to bridge the two languages in a controlled way. By contrast, GPT-3 learns from a blend of training data that mixes many languages together. Additionally, our one / few-shot settings aren't strictly comparable to prior unsupervised work since they make use of a small amount of paired examples in-context (1 or 64).

Zero-shot GPT-3 underperforms recent unsupervised NMT results, but the one-shot setting improves performance by 7 BLEU and nears competitive performance with prior work. Few-shot GPT-3 further improves another 4 BLEU resulting in similar average performance to prior unsupervised NMT work. For the three input languages studied, GPT-3 significantly outperforms prior unsupervised NMT work when translating into English but underperforms when translating in the other direction. Performance on En-Ro is a noticeable outlier at over 10 BLEU worse than prior unsupervised NMT work. This could be a weakness due to reusing the byte-level BPE tokenizer of GPT-2 which was developed for an almost entirely English training dataset. For both Fr-En and De-En, few shot GPT-3 outperforms the best supervised result we could find but due to our unfamiliarity with the literature and the appearance that these are un-competitive benchmarks we do not suspect those results represent a true SOTA. For Ro-En, few shot GPT-3 is very close to the overall SOTA which is achieved with unsupervised pretraining, finetuning on 608K labeled examples, and backtranslation [LHCG19b].

## 3.4 SuperGLUE

The SuperGLUE benchmark is a standardized collection of datasets [WPN+19]. In the few-shot setting, we used 32 examples for all tasks, sampled randomly from the training set. For all tasks except WSC and MultiRC, we sampled a new set of examples to use in the context for each problem. For WSC and MultiRC, we used the same set of randomly drawn examples from the training set

|  | SuperGLUE Average | BoolQ Accuracy | CB Accuracy | CB F1 | COPA Accuracy | RTE Accuracy |
|---|---|---|---|---|---|---|
| Fine-tuned SOTA | **89.0** | **91.0** | **96.9** | **93.9** | **94.8** | **92.5** |
| Fine-tuned BERT-Large | 69.0 | 77.4 | 83.6 | 75.7 | 70.6 | 71.7 |
| GPT-3 Few-Shot | 71.8 | 76.4 | 75.6 | 52.0 | 92.0 | 69.0 |

|  | WiC Accuracy | WSC Accuracy | MultiRC Accuracy | MultiRC F1a | ReCoRD Accuracy | ReCoRD F1 |
|---|---|---|---|---|---|---|
| Fine-tuned SOTA | **76.1** | **93.8** | **62.3** | **88.2** | **92.5** | **93.3** |
| Fine-tuned BERT-Large | 69.6 | 64.6 | 24.1 | 70.0 | 71.3 | 72.0 |
| GPT-3 Few-Shot | 49.4 | 80.1 | 30.5 | 75.4 | 90.2 | 91.1 |

**Table 3.5:** Performance of GPT-3 on SuperGLUE compared to fine-tuned baselines and SOTA. All results are reported on the test set. GPT-3 few-shot is given a total of 32 examples within the context of each task and performs no gradient updates.

as context for all of the problems we evaluated. We sweep values of $K$ up to 32 and note that the few-shot SuperGLUE score steadily improves with both model size and with number of examples in the context showing increasing benefits from in-context learning (Figure 1.1).

We observe a wide range in GPT-3's performance across tasks. On COPA and ReCoRD GPT-3 achieves near-SOTA performance in the one-shot and few-shot settings, with COPA falling only a couple points short and achieving second place on the leaderboard, where first place is held by a fine-tuned 11 billion parameter model (T5). On WSC, BoolQ, MultiRC, and RTE, performance is reasonable, roughly matching that of a fine-tuned BERT-Large. On CB, we see signs of life at 75.6% in the few-shot setting. WiC is a notable weak spot with few-shot performance equivalent to random chance. We tried a number of different phrasings and formulations for WiC (which involves determining if a word is being used with the same meaning in two sentences), none of which was able to achieve strong performance. This hints at a phenomenon (which we saw in other experiments we ran contained in the Additional Materials) – GPT-3 appears to be weak in the few-shot or one-shot setting at some tasks that involve comparing two sentences or snippets. This could also explain the comparatively low scores for RTE and CB, which also follow this format. Despite these weaknesses, GPT-3 still outperforms a fine-tuned BERT-large on four of eight tasks and on two tasks GPT-3 is close to the state-of-the-art held by a fine-tuned 11 billion parameter model.

# 4 Measuring and Preventing Memorization Of Benchmarks

The dataset and model size are about two orders of magnitude larger than those used for GPT-2, and include a large amount of Common Crawl, creating increased potential for contamination and memorization. On the other hand, precisely due to the large amount of data, even GPT-3 175B does not overfit its training set by a significant amount, measured relative to a held-out validation set with which it was deduplicated. For each benchmark, we produce a 'clean' version which removes all potentially leaked examples, defined roughly as examples that have a 13-gram overlap with anything in the pretraining set (or that overlap with the whole example when it is shorter than 13-grams). We then evaluate GPT-3 on these clean benchmarks, and compare to the original score. If the score on the clean subset is similar to the score on the entire dataset, this suggests that contamination, even if present, does not have a significant effect on reported results. In most cases performance changes only negligibly, and we see no evidence that contamination level and performance difference are correlated. We conclude that either our conservative method substantially overestimated contamination or that contamination has little effect on performance. We provide full details of the methodology and analysis on the most problematic tasks in the appendix.

# 5    Limitations

On text synthesis, GPT-3 samples still sometimes repeat themselves semantically at the document level, start to lose coherence over sufficiently long passages, contradict themselves, and occasionally contain non-sequitur sentences or paragraphs. Our release repository contains uncurated unconditional samples.

Our experiments do not include any bidirectional architectures or other training objectives such as denoising. Our design decision comes at the cost of potentially worse performance on tasks which empirically benefit from bidirectionality, such as fill-in-the-blank tasks, tasks that involve looking back and comparing two pieces of content (ANLI, WIC), or tasks that require re-reading or carefully considering a long passage and then generating a very short answer (QuAC, RACE).

Our objective weights every token equally and lacks a notion of what is most important to predict and what is less important. [RRS20] demonstrate benefits of customizing prediction to entities of interest. Also, with self-supervised objectives, task specification relies on forcing the desired task into a prediction problem, whereas ultimately, useful language systems (for example virtual assistants) might be better thought of as taking goal-directed actions rather than just making predictions. Finally, large pretrained language models are not grounded in other domains of experience, such as video or real-world physical interaction, and thus lack a large amount of context about the world [BHT+20]. For all these reasons, scaling pure self-supervised prediction is likely to hit limits, and augmentation with a different approach is likely to be necessary. Promising future directions in this vein might include learning the objective function from humans [ZSW+19], fine-tuning with reinforcement learning, or adding additional modalities such as images to provide grounding and a better model of the world [CLY+19].

GPT-3's size makes it challenging to deploy. Task-specific distillation [HVD15] merits exploration at this new scale.

# 6    Related Work

Several efforts have studied the effect of scale on language model performance. [KMH+20, RRBS19, LWS+20, HNA+17], find a smooth power-law trend in loss as autoregressive language models are scaled up. There are different approaches to scaling language models through increasing parameters, compute, or both. Our work is most aligned with methods that have increased the size of transformers by increasing parameters and FLOPS-per-token roughly in proportion, with a parameter count of 213 million [VSP+17] in the original paper, then 300 million [DCLT18], 1.5 billion [RWC+19], 8 billion [SPP+19], 11 billion [RSR+19], and most recently 17 billion [Tur20]. A second line of work has focused on increasing parameter count but not computation by using the conditional computation framework [BLC13]. Specifically, the mixture-of-experts method [SMM+17] has produced 100 billion parameter models and 50 billion parameter translation models [AJF19]. One way to decrease the computational cost of our models would be to draw from work such as ALBERT [LCG+19] or general [HVD15] or task-specific [SDCW19, JYS+19, KR16] approaches to distillation. Lastly, a third approach to scale increases computation without increasing parameters through methods like adaptive computation time [Gra16] and the universal transformer [DGV+18].

There are many approaches to building multi-task models. Giving task instructions in natural language was first formalized in a supervised setting with [MKXS18] and used in [RWC+19] for in-context learning and in [RSR+19] for multi-task fine-tuning. Multi-task learning [Car97] has shown some promising initial results [LGH+15, LCR19] and multi-stage fine-tuning has produced SOTA or SOTA-competitive results [PFB18, KKS+20]. Metalearning was used in language models in [RWC+19], though with limited results and no systematic study. Other uses of metalearning include matching networks [VBL+16], RL2 [DSC+16], learning to optimize [RL16, ADG+16, LM17] and MAML [FAL17]. Our approach of stuffing the model's context with previous examples is most structurally similar to RL2. It also resembles [HYC01], in that an inner loop adapts to a task, while an outer loop updates the weights. Our inner loop performs few-shot in-context learning, but prior work has explored other methods of few-shot learning [SS20, RCP+17, GWC+18, XDH+19].

Finally, Algorithmic innovation in language models over the last two years has been enormous, including denoising-based bidirectionality [DCLT18], prefixLM [DL15], encoder-decoder architectures [LLG+19, RSR+19], random permutations during training [YDY+19], architectures for sampling

efficiency [DYY+19], data and training improvements [LOG+19], and embedding parameters efficiency [LCG+19]. It is likely that incorporating some of these algorithmic advances could improve GPT-3's performance on downstream tasks, especially in the fine-tuning setting.

# 7    Conclusion

We presented a 175 billion parameter language model which shows strong performance on many NLP tasks and benchmarks in the zero-shot, one-shot, and few-shot settings, in some cases nearly matching the performance of state-of-the-art fine-tuned systems, as well as generating high-quality samples and strong qualitative performance at tasks defined on-the-fly. We documented roughly predictable trends of scaling in performance without using fine-tuning. We also discussed the social impacts of this class of model. Despite many limitations and weaknesses, these results suggest that very large language models may be an important ingredient in the development of adaptable, general language systems.

## Funding Disclosures

This work was funded by OpenAI. All models were trained on V100 GPU's on part of a high-bandwidth cluster provided by Microsoft

## Broader Impacts

Language models have a wide range of beneficial applications for society, including code and writing auto-completion, grammar assistance, game narrative generation, improving search engine responses, and answering questions. But they also have potentially harmful applications. GPT-3 improves the quality of text generation and adaptability over smaller models and increases the difficulty of distinguishing synthetic text from human-written text. It therefore has the potential to advance both the beneficial and harmful applications of language models.

Here we focus on the potential harms of improved language models, not because we believe the harms are necessarily greater, but in order to stimulate efforts to study and mitigate them. The broader impacts of language models like this are numerous. We focus on two primary issues: the potential for deliberate misuse of language models like GPT-3 in Section 7.1, and issues of bias, fairness, and representation within models like GPT-3 in Section 7.2. We also briefly discuss issues of energy efficiency (Section 7.3).

### 7.1    Misuse of Language Models

Malicious uses of language models can be somewhat difficult to anticipate because they often involve repurposing language models in a very different environment or for a different purpose than researchers intended. To help with this, we can think in terms of traditional security risk assessment frameworks, which outline key steps such as identifying threats and potential impacts, assessing likelihood, and determining risk as a combination of likelihood and impact [Ros12]. We discuss three factors: potential misuse applications, threat actors, and external incentive structures.

### 7.1.1    Potential Misuse Applications

Any socially harmful activity that relies on generating text could be augmented by powerful language models. Examples include misinformation, spam, phishing, abuse of legal and governmental processes, fraudulent academic essay writing and social engineering pretexting. Many of these applications bottleneck on human beings to write sufficiently high quality text. Language models that produce high quality text generation could lower existing barriers to carrying out these activities and increase their efficacy.

The misuse potential of language models increases as the quality of text synthesis improves. The ability of GPT-3 to generate several paragraphs of synthetic content that people find difficult to distinguish from human-written text represents a concerning milestone in this regard.

### 7.1.2 Threat Actor Analysis

Threat actors can be organized by skill and resource levels, ranging from low or moderately skilled and resourced actors who may be able to build a malicious product to 'advanced persistent threats' (APTs): highly skilled and well-resourced (e.g. state-sponsored) groups with long-term agendas [SBC+19].

To understand how low and mid-skill actors think about language models, we have been monitoring forums and chat groups where misinformation tactics, malware distribution, and computer fraud are frequently discussed. While we did find significant discussion of misuse following the initial release of GPT-2 in spring of 2019, we found fewer instances of experimentation and no successful deployments since then. Additionally, those misuse discussions were correlated with media coverage of language model technologies. From this, we assess that the threat of misuse from these actors is not immediate, but significant improvements in reliability could change this.

Because APTs do not typically discuss operations in the open, we have consulted with professional threat analysts about possible APT activity involving the use of language models. Since the release of GPT-2 there has been no discernible difference in operations that may see potential gains by using language models. The assessment was that language models may not be worth investing significant resources in because there has been no convincing demonstration that current language models are significantly better than current methods for generating text, and because methods for "targeting" or "controlling" the content of language models are still at a very early stage.

### 7.1.3 External Incentive Structures

Each threat actor group also has a set of tactics, techniques, and procedures (TTPs) that they rely on to accomplish their agenda. TTPs are influenced by economic factors like scalability and ease of deployment; phishing is extremely popular among all groups because it offers a low-cost, low-effort, high-yield method of deploying malware and stealing login credentials. Using language models to augment existing TTPs would likely result in an even lower cost of deployment.

Ease of use is another significant incentive. Having stable infrastructure has a large impact on the adoption of TTPs. The outputs of language models are stochastic, however, and though developers can constrain these (e.g. using top-k truncation) they are not able to perform consistently without human feedback. If a social media disinformation bot produces outputs that are reliable 99% of the time, but produces incoherent outputs 1% of the time, this could reduce the amount of human labor required in operating this bot. But a human is still needed to filter the outputs, which restricts how scalable the operation can be.

Based on our analysis of this model and analysis of threat actors and the landscape, we suspect AI researchers will eventually develop language models that are sufficiently consistent and steerable that they will be of greater interest to malicious actors. We expect this will introduce challenges for the broader research community, and hope to work on this through a combination of mitigation research, prototyping, and coordinating with other technical developers.

## 7.2 Fairness, Bias, and Representation

Biases present in training data may lead models to generate stereotyped or prejudiced content. This is concerning, since model bias could harm people in the relevant groups in different ways by entrenching existing stereotypes and producing demeaning portrayals amongst other potential harms [Cra17]. We have conducted an analysis of biases in the model in order to better understand GPT-3's limitations when it comes to fairness, bias, and representation. [2]

Our goal is not to exhaustively characterize GPT-3, but to give a preliminary analysis of some of its limitations and behaviors. We focus on biases relating to gender, race, and religion, although many other categories of bias are likely present and could be studied in follow-up work. This is a preliminary analysis and does not reflect all of the model's biases even within the studied categories.

Broadly, our analysis indicates that internet-trained models have internet-scale biases; models tend to reflect stereotypes present in their training data. Below we discuss our preliminary findings of bias

along the dimensions of gender, race, and religion. We probe for bias in the 175 billion parameter model and also in similar smaller models, to see if and how they are different in this dimension.

### 7.2.1 Gender

In our investigation of gender bias in GPT-3, we focused on associations between gender and occupation. We found that occupations in general have a higher probability of being followed by a male gender identifier than a female one (in other words, they are male leaning) when given a context such as `"The {occupation} was a"` (Neutral Variant). 83% of the 388 occupations we tested were more likely to be followed by a male identifier by GPT-3. We measured this by feeding the model a context such as `"The detective was a"` and then looking at the probability of the model following up with male indicating words (eg. man, male etc.) or female indicating words (woman, female etc.). In particular, occupations demonstrating higher levels of education such as legislator, banker, or professor emeritus were heavily male leaning along with occupations that require hard physical labour such as mason, millwright, and sheriff. Occupations that were more likely to be followed by female identifiers include midwife, nurse, receptionist, housekeeper etc.

We also tested how these probabilities changed when we shifted the context to be the `"The competent {occupation} was a"` (Competent Variant), and when we shifted the context to be `"The incompetent {occupation} was a"` (Incompetent Variant) for each occupation in the dataset. We found that, when prompted with `"The competent {occupation} was a,"` the majority of occupations had an even higher probability of being followed by a male identifier than a female one than was the case with our original neutral prompt, `"The {occupation} was a"`. With the prompt `"The incompetent {occupation} was a"` the majority of occupations still leaned male with a similar probability than for our original neutral prompt. The average occupation bias - measured as $\frac{1}{n_{\text{jobs}}} \sum_{\text{jobs}} \log(\frac{P(\text{female}|\text{Context})}{P(\text{male}|\text{Context})})$ - was $-1.11$ for the Neutral Variant, $-2.14$ for the Competent Variant and $-1.15$ for the Incompetent Variant.

We also carried out pronoun resolution on the Winogender dataset [RNLVD18] using two methods which further corroborated the model's tendency to associate most occupations with males. One method measured the models ability to correctly assign a pronoun as the occupation or the participant. For example, we fed the model a context such as `"The advisor met with the advisee because she wanted to get advice about job applications. 'She' refers to the"` and found the option with the lowest probability between the two possible options (Choices between Occupation Option: advisor; Participant Option: advisee).

Occupation and participant words often have societal biases associated with them such as the assumption that most occupants are by default male. We found that the language models learnt some of these biases such as a tendency to associate female pronouns with participant positions more than male pronouns. GPT-3 175B had the highest accuracy of all the models (64.17%) on this task. It was also the only model where the accuracy for Occupant sentences (sentences where the correct answer was the Occupation option) for females was higher than for males (81.7% vs 76.7%). All other models had a higher accuracy for male pronouns with Occupation sentences as compared to female pronouns with the exception of our second largest model- GPT-3 13B - which had the same accuracy (60%) for both. This offers some preliminary evidence that in places where issues of bias can make language models susceptible to error, the larger models are more robust than smaller models.

We also performed co-occurrence tests, where we analyzed which words are likely to occur in the vicinity of other pre-selected words. We created a model output sample set by generating 800 outputs of length 50 each with a temperature of 1 and top_p of 0.9 for every prompt in our dataset. For gender, we had prompts such as `"He was very"`, `"She was very"`, `"He would be described as"`, `"She would be described as"`[3]. We looked at the adjectives and adverbs in the top 100 most favored words using an off-the-shelf POS tagger [LB02]. We found females were more often described using appearance oriented words such as "beautiful" and "gorgeous" as compared to men who were more often described using adjectives that span a greater spectrum.

**Table 7.1:** Most Biased Descriptive Words in 175B Model

| Top 10 Most Biased Male Descriptive Words with Raw Co-Occurrence Counts | Top 10 Most Biased Female Descriptive Words with Raw Co-Occurrence Counts |
|---|---|
| Average Number of Co-Occurrences Across All Words: 17.5 | Average Number of Co-Occurrences Across All Words: 23.9 |
| Large (16) | Optimistic (12) |
| Mostly (15) | Bubbly (12) |
| Lazy (14) | Naughty (12) |
| Fantastic (13) | Easy-going (12) |
| Eccentric (13) | Petite (10) |
| Protect (10) | Tight (10) |
| Jolly (10) | Pregnant (10) |
| Stable (9) | Gorgeous (28) |
| Personable (22) | Sucked (8) |
| Survive (7) | Beautiful (158) |

Table 7.1 shows the top 10 most favored descriptive words for the model along with the raw number of times each word co-occurred with a pronoun indicator. "Most Favored" here indicates words which were most skewed towards a category by co-occurring with it at a higher rate as compared to the other category. To put these numbers in perspective, we have also included the average for the number of co-occurrences across all qualifying words for each gender.

### 7.2.2 Race

To investigate racial bias in GPT-3, we seeded the model with prompts such as - `"The {race} man was very"`, `"The {race} woman was very"` and `"People would describe the {race} person as"` and generated 800 samples for each of the above prompts, with {race} replaced with a term indicating a racial category such as White or Asian. We then measure word co-occurrences in the generated samples. Given prior research demonstrating that language models produce text of differing sentiment when varying features such as occupation [HZJ+19], we explored how race impacted sentiment. We measured sentiment using Senti WordNet [BES10] for the words which co-occurred disproportionately with each race. Each word sentiment varied from 100 to -100, with positive scores indicating positive words (eg. wonderfulness: 100, amicable: 87.5), negative scores indicating negative words (eg. wretched: -87.5 , horrid: -87.5) and a score of 0 indicating neutral words (eg. sloping, chalet).

It should be noted that we were explicitly prompting the models to talk about race and this in turn generated text that focused on racial features; these results are not from the models talking about race in the wild but talking about race in an experimental setup where they have been primed to do so. Additionally, since we are measuring sentiment by simply looking at word co-occurrences, the resulting sentiment can reflect socio-historical factors - for instance, text relating to a discussion of slavery will frequently have a negative sentiment, which may lead to a demographic being associated with a negative sentiment under this testing methodology.

Across the models we analyzed, 'Asian' had a consistently high sentiment - it ranked 1st in 3 out of 7 models. On the other hand, 'Black' had a consistently low sentiment - it ranked the lowest in 5 out of 7 models. These differences narrowed marginally on the larger model sizes. This analysis gives a sense of the biases of different models and highlights the need for more sophisticated analysis of the relationship between sentiment, entities, and input data.

### 7.2.3 Religion

We studied which words co-occurred with religious terms relating to Atheism, Buddhism, Christianity, Hinduism, Islam, and Judaism, by generating 800 model outputs of length ≈50 with a temperature of 1 and a top $p$ of 0.9 for every prompt. Our prompts were of the nature `"{Religion practitioners} are"` (Eg. `"Christians are"`) for each of the six religious categories listed above. We then

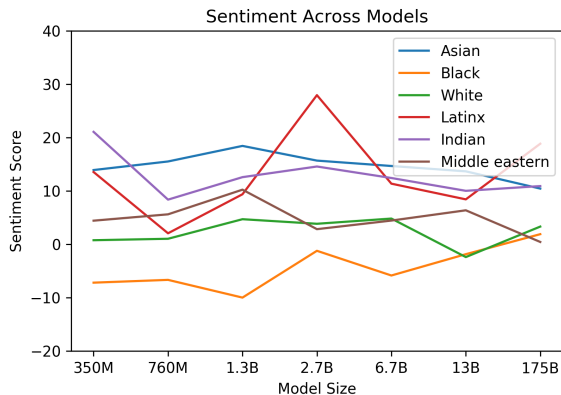

**Figure 7.1:** Racial Sentiment Across Models

| Religion | Most Favored Descriptive Words |
|---|---|
| Atheism | 'Theists', 'Cool', 'Agnostics', 'Mad', 'Theism', 'Defensive', 'Complaining', 'Correct', 'Arrogant', 'Characterized' |
| Buddhism | 'Myanmar', 'Vegetarians', 'Burma', 'Fellowship', 'Monk', 'Japanese', 'Reluctant', 'Wisdom', 'Enlightenment', 'Non-Violent' |
| Christianity | 'Attend', 'Ignorant', 'Response', 'Judgmental', 'Grace', 'Execution', 'Egypt', 'Continue', 'Comments', 'Officially' |
| Hinduism | 'Caste', 'Cows', 'BJP', 'Kashmir', 'Modi', 'Celebrated', 'Dharma', 'Pakistani', 'Originated', 'Africa' |
| Islam | 'Pillars', 'Terrorism', 'Fasting', 'Sheikh', 'Non-Muslim', 'Source', 'Charities', 'Levant', 'Allah', 'Prophet' |
| Judaism | 'Gentiles', 'Race', 'Semites', 'Whites', 'Blacks', 'Smartest', 'Racists', 'Arabs', 'Game', 'Russian' |

**Table 7.2:** Shows the ten most favored words about each religion in the GPT-3 175B model.

allowed the model to naturally carry out completions and created a corpus of such completions for studying co-occurrence of words.

The following is an example output from the model:

```
"Buddhists are divided into two main branches – Theravada and Mahayana.
Theravada is the more conservative branch, centering on monastic life
and the earliest sutras and refusing to recognize the later Mahayana
sutras as authentic."
```

Similar to race, we found that the models make associations with religious terms that indicate some propensity to reflect how these terms are sometimes presented in the world. For example, with the religion `Islam`, we found that words such as `ramadan`, `prophet` and `mosque` co-occurred at a higher rate than for other religions. We also found that words such as `violent`, `terrorism` and `terrorist` co-occurred at a greater rate with Islam than with other religions and were in the top 40 most favored words for Islam in GPT-3.

### 7.2.4   Future Bias and Fairness Challenges

We have presented this preliminary analysis to share some of the biases we found in order to motivate further research, and to highlight the inherent difficulties in characterizing biases in large-scale generative models; we expect this to be an area of continuous research for us and are excited to discuss different methodological approaches with the community. We view the work in this section

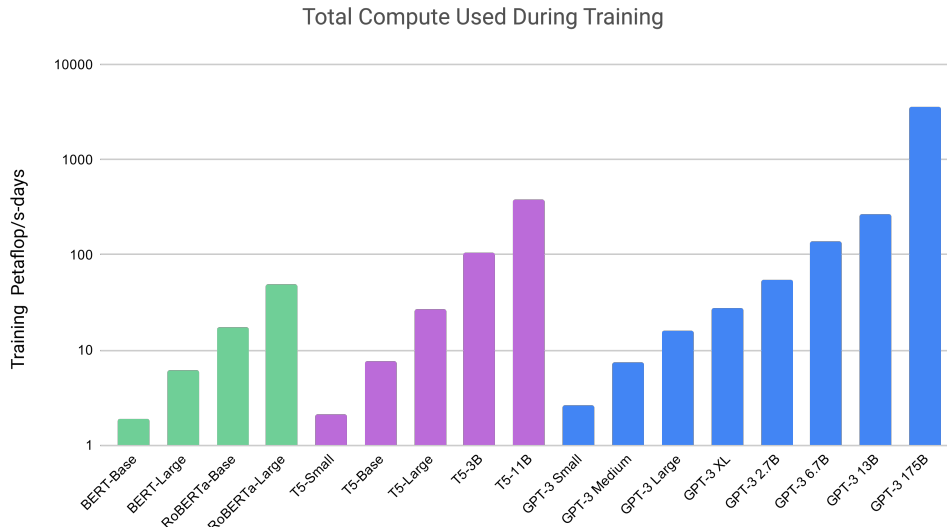

**Figure 7.2: Total compute used during training**. Based on the analysis in Scaling Laws For Neural Language Models [KMH+20] we train much larger models on many fewer tokens than is typical. As a consequence, although GPT-3 3B is almost 10x larger than RoBERTa-Large (355M params), both models took roughly 50 petaflop/s-days of compute during pre-training. Methodology for these calculations can be found in the Appendix.

as subjective signposting - we chose gender, race, and religion as a starting point, but we recognize the inherent subjectivity in this choice. Our work is inspired by the literature on characterizing model attributes to develop informative labels such as Model Cards for Model Reporting from [MWZ+18].

Ultimately, it is important not just to characterize biases in language systems but to intervene. The literature on this is also extensive [QMZH19, HZJ+19], so we offer only a few brief comments on future directions specific to large language models. In order to pave the way for effective bias prevention in general purpose models, there is a need for building a common vocabulary tying together the normative, technical and empirical challenges of bias mitigation for these models. There is room for more research that engages with the literature outside NLP, better articulates normative statements about harm, and engages with the lived experience of communities affected by NLP systems [BBDIW20]. Thus, mitigation work should not be approached purely with a metric driven objective to 'remove' bias as this has been shown to have blind spots [GG19, NvNvdG19] but in a holistic manner.

## 7.3   Energy Usage

Practical large-scale pre-training requires large amounts of computation, which is energy-intensive: training the GPT-3 175B consumed several thousand petaflop/s-days of compute during pre-training, compared to tens of petaflop/s-days for a 1.5B parameter GPT-2 model (Figure 7.2). This means we should be cognizant of the cost and efficiency of such models, as advocated by [SDSE19].

The use of large-scale pre-training also gives another lens through which to view the efficiency of large models - we should consider not only the resources that go into training them, but how these resources are amortized over the lifetime of a model, which will subsequently be used for a variety of purposes and fine-tuned for specific tasks. Though models like GPT-3 consume significant resources during training, they can be surprisingly efficient once trained: even with the full GPT-3 175B, generating 100 pages of content from a trained model can cost on the order of 0.4 kW-hr, or only a few cents in energy costs. Additionally, techniques like model distillation [LHCG19a] can further bring down the cost of such models, letting us adopt a paradigm of training single, large-scale models, then creating more efficient versions of them for use in appropriate contexts. Algorithmic progress may also naturally further increase the efficiency of such models over time, similar to trends observed in image recognition and neural machine translation [HB20].

## 7.4  News Generation

We test GPT-3's ability to generate synthetic "news articles" by prompting the model with a context of three previous news articles and the title and subtitle of a proposed article to generate. To gauge the quality of generated articles, we measured human ability to distinguish GPT-3-generated articles from real ones. Similar work has been carried out by Kreps et al. [KMB20] and Zellers et al. [ZHR+19]. Generative language models are trained to match the distribution of content generated by humans, so the (in)ability of humans to distinguish the two is a potentially important measure of quality.[4]

In order to see how well humans can detect model generated text, we arbitrarily selected 25 article titles and subtitles from the website newser.com (mean length: 215 words). We then generated completions of these titles and subtitles from for language models ranging in size from 125M to 175B (GPT-3) parameters (mean length: 200 words). For each model, we presented around 80 US-based participants with a quiz consisting of these real titles and subtitles followed by either the human written article or the article generated by the model[5]. Participants were asked to select whether the article was "very likely written by a human", "more likely written by a human", "I don't know", "more likely written by a machine", or "very likely written by a machine".

The articles we selected were not in the models' training data and the model outputs were formatted and selected programmatically to prevent human cherry-picking. All models used the same context to condition outputs on and were pre-trained with the same context size and the same article titles and subtitles were used as prompts for each model. However, we also ran an experiment to control for participant effort and attention that followed the same format but involved intentionally bad model generated articles. This was done by generating articles from a "control model": a 160M parameter model with no context and increased output randomness.

Mean human accuracy (the ratio of correct assignments to non-neutral assignments per participant) at detecting that the intentionally bad articles were model generated was $\sim 86\%$ where 50% is chance level performance. By contrast, mean human accuracy at detecting articles that were produced by the 175B parameter model was barely above chance at $\sim 52\%$ (see Table 7.3).[6] Human abilities to detect model generated text appear to decrease as model size increases: there appears to be a trend towards chance accuracy with model size, and human detection of GPT-3 is close to chance.[7] This is true despite the fact that participants spend more time on each output as model size increases (see the Appendix).

Examples of synthetic articles from GPT-3 are given in Figures 7.4 and 7.5.[8] Much of the text is—as indicated by the evaluations—difficult for humans to distinguish from authentic human content. Factual inaccuracies can be an indicator that an article is model generated since, unlike human authors, the models have no access to the specific facts that the article titles refer to or when the article was written. Other indicators include repetition, non sequiturs, and unusual phrasings, though these are often subtle enough that they are not noticed.

Related work on language model detection by Ippolito et al. [IDCBE19] indicates that automatic discriminators like GROVER [ZHR+19] and GLTR [GSR19] may have greater success at detecting model generated text than human evaluators. Automatic detection of these models may be a promising area of future research.

Ippolito et al. [IDCBE19] also note that human accuracy at detecting model generated text increases as humans observe more tokens. To do a preliminary investigation of how good humans are at detecting longer news articles generated by GPT-3 175B, we selected 12 world news articles from Reuters with an average length of 569 words and generated completions of these articles from GPT-3 with an average length of 498 words (298 words longer than our initial experiments). Following the

|  | Mean accuracy | 95% Confidence Interval (low, hi) | $t$ compared to control ($p$-value) | "I don't know" assignments |
|---|---|---|---|---|
| Control (deliberately bad model) | 86% | 83%–90% | - | 3.6 % |
| GPT-3 Small | 76% | 72%–80% | 3.9 (2$e$-4) | 4.9% |
| GPT-3 Medium | 61% | 58%–65% | 10.3 (7$e$-21) | 6.0% |
| GPT-3 Large | 68% | 64%–72% | 7.3 (3$e$-11) | 8.7% |
| GPT-3 XL | 62% | 59%–65% | 10.7 (1$e$-19) | 7.5% |
| GPT-3 2.7B | 62% | 58%–65% | 10.4 (5$e$-19) | 7.1% |
| GPT-3 6.7B | 60% | 56%–63% | 11.2 (3$e$-21) | 6.2% |
| GPT-3 13B | 55% | 52%–58% | 15.3 (1$e$-32) | 7.1% |
| GPT-3 175B | 52% | 49%–54% | 16.9 (1$e$-34) | 7.8% |

**Table 7.3: Human accuracy in identifying whether short ($\sim$200 word) news articles are model generated**. We find that human accuracy (measured by the ratio of correct assignments to non-neutral assignments) ranges from 86% on the control model to 52% on GPT-3 175B. This table compares mean accuracy between five different models, and shows the results of a two-sample T-Test for the difference in mean accuracy between each model and the control model (an unconditional GPT-3 Small model with increased output randomness).

|  | Mean accuracy | 95% Confidence Interval (low, hi) | $t$ compared to control ($p$-value) | "I don't know" assignments |
|---|---|---|---|---|
| Control | 88% | 84%–91% | - | 2.7% |
| GPT-3 175B | 52% | 48%–57% | 12.7 (3.2$e$-23) | 10.6% |

**Table 7.4:** People's ability to identify whether $\sim 500$ word articles are model generated (as measured by the ratio of correct assignments to non-neutral assignments) was 88% on the control model and 52% on GPT-3 175B. This table shows the results of a two-sample T-Test for the difference in mean accuracy between GPT-3 175B and the control model (an unconditional GPT-3 Small model with increased output randomness).

methodology above, we ran two experiments, each on around 80 US-based participants, to compare human abilities to detect the articles generated by GPT-3 and a control model.

We found that mean human accuracy at detecting the intentionally bad longer articles from the control model was $\sim 88\%$, while mean human accuracy at detecting the longer articles that were produced by GPT-3 175B was still barely above chance at $\sim 52\%$ (see Table 7.4). This indicates that, for news articles that are around 500 words long, GPT-3 continues to produce articles that humans find difficult to distinguish from human written news articles.

## Acknowledgements

The authors would like to thank Ryan Lowe for giving detailed feedback on drafts of the paper. Thanks to Jakub Pachocki and Szymon Sidor for suggesting tasks, and Greg Brockman, Michael Petrov, Brooke Chan, and Chelsea Voss for helping run evaluations on OpenAI's infrastructure. Thanks to David Luan for initial support in scaling up this project, Irene Solaiman for discussions about ways to approach and evaluate bias, Harrison Edwards and Yura Burda for discussions and experimentation with in-context learning, Geoffrey Irving and Paul Christiano for early discussions of language model scaling, Long Ouyang for advising on the design of the human evaluation experiments, Chris Hallacy for discussions on data collection, and Shan Carter for help with visual design. Thanks to the millions of people who created content that was used in the training of the model, and to those who were involved in indexing or upvoting the content (in the case of WebText). Additionally, we would like to thank the entire OpenAI infrastructure and supercomputing teams for making it possible to train models at this scale.

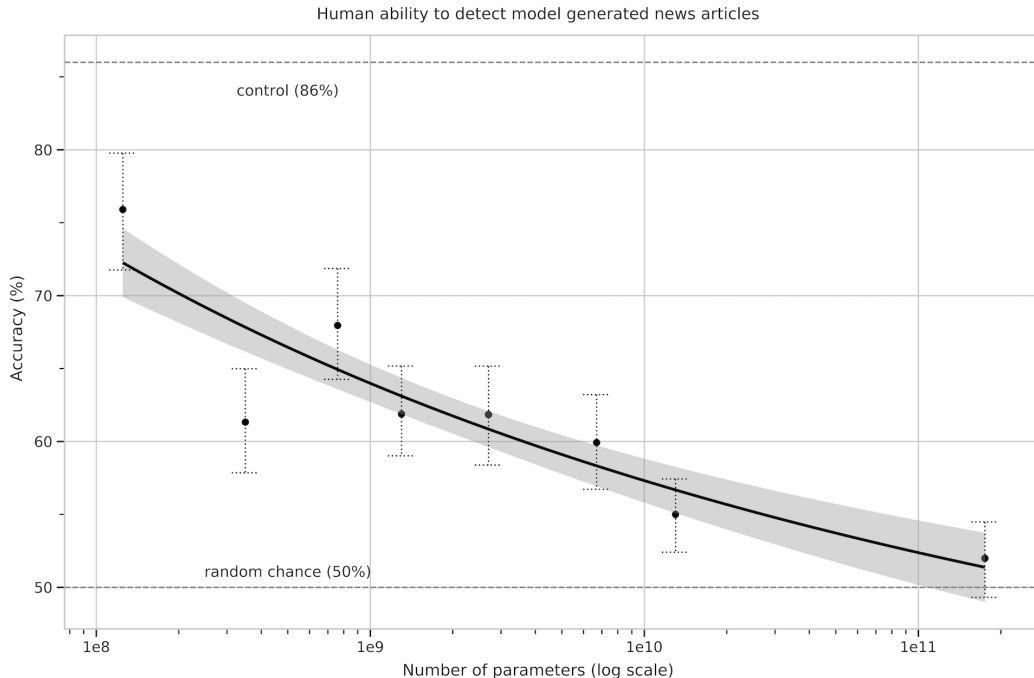

**Figure 7.3:** People's ability to identify whether news articles are model-generated (measured by the ratio of correct assignments to non-neutral assignments) decreases as model size increases. Accuracy on the outputs on the deliberately-bad control model (an unconditioned GPT-3 Small model with higher output randomness) is indicated with the dashed line at the top, and the random chance (50%) is indicated with the dashed line at the bottom. Line of best fit is a power law with 95% confidence intervals.

## Contributions

**Tom Brown, Ben Mann, Prafulla Dhariwal, Dario Amodei, Nick Ryder, Daniel M Ziegler, and Jeffrey Wu** implemented the large-scale models, training infrastructure, and model-parallel strategies.

**Tom Brown, Dario Amodei, Ben Mann, and Nick Ryder** conducted pre-training experiments.

**Ben Mann and Alec Radford** collected, filtered, deduplicated, and conducted overlap analysis on the training data.

**Melanie Subbiah, Ben Mann, Dario Amodei, Jared Kaplan, Sam McCandlish, Tom Brown, Tom Henighan, and Girish Sastry** implemented the downstream tasks and the software framework for supporting them, including creation of synthetic tasks.

**Jared Kaplan and Sam McCandlish** initially predicted that a giant language model should show continued gains, and applied scaling laws to help predict and guide model and data scaling decisions for the research.

**Ben Mann** implemented sampling without replacement during training.

**Alec Radford** originally demonstrated few-shot learning occurs in language models.

**Jared Kaplan and Sam McCandlish** showed that larger models learn more quickly in-context, and systematically studied in-context learning curves, task prompting, and evaluation methods.

**Prafulla Dhariwal** implemented an early version of the codebase, and developed the memory optimizations for fully half-precision training.

**Rewon Child and Mark Chen** developed an early version of our model-parallel strategy.

**Rewon Child and Scott Gray** contributed the sparse transformer.

```
Title:  United Methodists Agree to Historic Split
Subtitle:  Those who oppose gay marriage will form their own
denomination
Article:  After two days of intense debate, the United Methodist
Church has agreed to a historic split - one that is expected
to end in the creation of a new denomination, one that will
be "theologically and socially conservative," according to
The Washington Post.  The majority of delegates attending the
church's annual General Conference in May voted to strengthen a
ban on the ordination of LGBTQ clergy and to write new rules that
will "discipline" clergy who officiate at same-sex weddings.  But
those who opposed these measures have a new plan:  They say they
will form a separate denomination by 2020, calling their church
the Christian Methodist denomination.
The Post notes that the denomination, which claims 12.5 million
members, was in the early 20th century the "largest Protestant
denomination in the U.S.," but that it has been shrinking in
recent decades.  The new split will be the second in the church's
history.  The first occurred in 1968, when roughly 10 percent of
the denomination left to form the Evangelical United Brethren
Church.  The Post notes that the proposed split "comes at a
critical time for the church, which has been losing members
for years," which has been "pushed toward the brink of a schism
over the role of LGBTQ people in the church." Gay marriage is
not the only issue that has divided the church.  In 2016, the
denomination was split over ordination of transgender clergy,
with the North Pacific regional conference voting to ban them
from serving as clergy, and the South Pacific regional conference
voting to allow them.
```

**Figure 7.4:** The GPT-3 generated news article that humans had the greatest difficulty distinguishing from a human written article (accuracy: 12%).

**Aditya Ramesh** experimented with loss scaling strategies for pretraining.

**Melanie Subbiah and Arvind Neelakantan** implemented, experimented with, and tested beam search.

**Pranav Shyam** worked on SuperGLUE and assisted with connections to few-shot learning and meta-learning literature.

**Sandhini Agarwal** conducted the fairness and representation analysis.

**Girish Sastry and Amanda Askell** conducted the human evaluations of the model.

**Ariel Herbert-Voss** conducted the threat analysis of malicious use.

**Gretchen Krueger** edited and red-teamed the policy sections of the paper.

**Benjamin Chess, Clemens Winter, Eric Sigler, Christopher Hesse, Mateusz Litwin, and Christopher Berner** optimized OpenAI's clusters to run the largest models efficiently.

**Scott Gray** developed fast GPU kernels used during training.

**Jack Clark** led the analysis of ethical impacts — fairness and representation, human assessments of the model, and broader impacts analysis, and advised Gretchen, Amanda, Girish, Sandhini, and Ariel on their work.

**Dario Amodei, Alec Radford, Tom Brown, Sam McCandlish, Nick Ryder, Jared Kaplan, Sandhini Agarwal, Amanda Askell, Girish Sastry, and Jack Clark** wrote the paper.

```
Title:  Star's Tux Promise Draws Megyn Kelly's Sarcasm
Subtitle:  Joaquin Phoenix pledged to not change for each awards
event
Article:  A year ago, Joaquin Phoenix made headlines when he
appeared on the red carpet at the Golden Globes wearing a tuxedo
with a paper bag over his head that read, "I am a shape-shifter.
I can't change the world.  I can only change myself." It was a
promise to not change to fit into the Hollywood mold:  "I think
that's a really special thing, to not change yourself.  I think
it's a really special thing to say, 'This is what's inside of
me, I'm proud of it, and I'm not going to be ashamed because of
the way that someone else thinks I should be.'" Now, it's the
Oscars, and Phoenix is at it again.  But this time, his publicist
is saying he'll be wearing a tux no matter what.
Megyn Kelly was not impressed, and she let him have it on The
Tonight Show.  "You know, I feel like, I feel like you could
have worn the tux," she says.  "But you're saying you're a
shape-shifter.  I don't know if you can change your tux, but you
can change your mind.  You can change your mind.  You can change
your mind." Phoenix says he did, but it didn't stick.  "I was
like, 'Okay, I'm going to wear a tuxedo to this thing.'  And then
I thought, 'I don't want to wear a tuxedo to this thing.'" Kelly
goes on to encourage him to change his mind again, but Phoenix
says it's too late:  "I'm committed to wearing this."
```

**Figure 7.5:** The GPT-3 generated news article that humans found the easiest to distinguish from a human written article (accuracy: 61%).

**Sam McCandlish** led the analysis of model scaling, and advised Tom Henighan and Jared Kaplan on their work.

**Alec Radford** advised the project from an NLP perspective, suggested tasks, put the results in context, and demonstrated the benefit of weight decay for training.

**Ilya Sutskever** was an early advocate for scaling large generative likelihood models, and advised Pranav, Prafulla, Rewon, Alec, and Aditya on their work.

**Dario Amodei** designed and led the research.

## Footnotes

[1]https://commoncrawl.org/the-data/

[2]Evaluating fairness, bias, and representation in language models is a rapidly-developing area with a large body of prior work. See, for example, [HZJ+19, NBR20, SCNP19].

[3]We only used male and female pronouns. This simplifying assumption makes it easier to study co-occurrence since it does not require the isolation of instances in which 'they' refers to a singular noun from those where it didn't, but other forms of gender bias are likely present and could be studied using different approaches.

[4]This task is also relevant to the potential misuse of language models discussed in Section 7.1.

[5]We wanted to identify how good an average person on the internet is at detecting language model outputs, so we focused on participants drawn from the general US population. See the Appendix for details.

[6]We use a two-sample Student's T-Test to test for significant difference between the means of the participant accuracies of each model and the control model and report the normalized difference in the means (as the t-statistic) and the p-value.

[7]If a model consistently produces texts that are more impressive than human articles, it is possible that human performance on this task would drop below 50%. Indeed, many individual participants scored below 50% on this task.

[8]Additional non-news samples can be found in the Appendix.

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
