[Supplementary Material]

# A  Details of Common Crawl Filtering

We employed two techniques to improve the quality of the Common Crawl dataset: (1) filtering Common Crawl and (2) fuzzy deduplication:

1. In order to improve the quality of Common Crawl, we developed an automatic filtering method to remove low quality documents. Using the original WebText as a proxy for high-quality documents, we trained a classifier to distinguish these from raw Common Crawl. We then used this classifier to re-sample Common Crawl by prioritizing documents which were predicted by the classifier to be higher quality. The classifier is trained using logistic regression classifier with features from Spark's standard tokenizer and HashingTF [1]. For the positive examples, we used a collection of curated datasets such as WebText, Wikiedia, and our web books corpus as the positive examples, and for the negative examples, we used unfiltered Common Crawl. We used this classifier to score Common Crawl documents. We kept each document in our dataset iff

$$\text{np.random.pareto}(\alpha) > 1 - \text{document\_score}$$

   We chose $\alpha = 9$ in order to take mostly documents the classifier scored highly, but still include some documents that were out of distribution. $\alpha$ was chosen to match the distribution of scores from our classifier on WebText. We found this re-weighting increased quality as measured by loss on a range of out-of-distribution generative text samples.

2. To further improve model quality and prevent overfitting (which becomes increasingly important as model capacity increases), we fuzzily deduplicated documents (i.e. removed documents with high overlap with other documents) within each dataset using Spark's MinHashLSH implementation with 10 hashes, using the same features as were used for classification above. We also fuzzily removed WebText from Common Crawl. Overall this decreased dataset size by an average of 10%.

After filtering for duplicates and quality, we also partially removed text occurring in benchmark datasets, described in Appendix C.

# B  Details of Model Training

To train all versions of GPT-3, we use Adam with $\beta_1 = 0.9$, $\beta_2 = 0.95$, and $\epsilon = 10^{-8}$, we clip the global norm of the gradient at 1.0, and we use cosine decay for learning rate down to 10% of its value, over 260 billion tokens (after 260 billion tokens, training continues at 10% of the original learning rate). There is a linear LR warmup over the first 375 million tokens. We also gradually increase the batch size linearly from a small value (32k tokens) to the full value over the first 4-12 billion tokens of training, depending on the model size. Data are sampled without replacement during training (until an epoch boundary is reached) to minimize overfitting. All models use weight decay of 0.1 to provide a small amount of regularization [LH17].

During training we always train on sequences of the full $n_{\text{ctx}} = 2048$ token context window, packing multiple documents into a single sequence when documents are shorter than 2048, in order to increase computational efficiency. Sequences with multiple documents are not masked in any special way but instead documents within a sequence are delimited with a special end of text token, giving the language model the information necessary to infer that context separated by the end of text token is unrelated. This allows for efficient training without need for any special sequence-specific masking.

$n_{\text{params}}$ is the total number of trainable parameters, $n_{\text{layers}}$ is the total number of layers, $d_{\text{model}}$ is the number of units in each bottleneck layer (we always have the feedforward layer four times the size of the bottleneck layer, $d_{\text{ff}} = 4 * d_{\text{model}}$), and $d_{\text{head}}$ is the dimension of each attention head. All models use a context window of $n_{\text{ctx}} = 2048$ tokens.

**Figure C.1: GPT-3 Training Curves**  We measure model performance during training on a dedu-
plicated validation split of our training distribution. Though there is some gap between training and
validation performance, the gap grows only minimally with model size and training time, suggesting
that most of the gap comes from a difference in difficulty rather than overfitting.

## C    Details of Test Set Contamination Studies

Since our training dataset is sourced from the internet, it is possible that our model was trained on
some of our benchmark test sets. Accurately detecting test contamination from internet-scale datasets
is a new area of research without established best practices. While it is common practice to train
large models without investigating contamination, given the increasing scale of pretraining datasets,
we believe this issue is becoming increasingly important to attend to.

This concern is not just hypothetical. One of the first papers to train a language model on Common
Crawl data [TL18] detected and removed a training document which overlapped with one of their
evaluation datasets. Other work such as GPT-2 [RWC+19] also conducted post-hoc overlap analysis.
Their study was encouraging, finding that although models did perform moderately better on data
that overlapped between training and testing, this did not significantly impact reported results due to
the small fraction of data which was contaminated (often only a few percent).

GPT-3 operates in a somewhat different regime. On the one hand, the dataset and model size are about
two orders of magnitude larger than those used for GPT-2, and include a large amount of Common
Crawl, increasing the potential for contamination and memorization. On the other hand, precisely due
to the large amount of data, even GPT-3 175B does not overfit its training set by a significant amount,
measured relative to a held-out validation set with which it was deduplicated (Figure C.1). Thus, we
expect that contamination is likely to be frequent, but that its effects may not be as large as feared.

We initially tried to address the issue of contamination by proactively searching for and attempting to
remove any overlap between our training data and the development and test sets of all benchmarks
studied in this paper. Unfortunately, a bug resulted in only partial removal of all detected overlaps
from the training data. Due to the cost of training, it wasn't feasible to retrain the model. To address
this, we investigate in detail how the remaining detected overlap impacts results.

For each benchmark, we produce a 'clean' version which removes all potentially leaked examples,
defined roughly as examples that have a 13-gram overlap with anything in the pretraining set (or that
overlap with the whole example when it is shorter than 13-grams). The goal is to very conservatively
flag anything that could potentially be contamination, so as to produce a clean subset that is free of
contamination with high confidence.

We then evaluate GPT-3 on these clean benchmarks, and compare to the original score. If the score
on the clean subset is similar to the score on the entire dataset, this suggests that contamination, even
if present, does not have a significant effect on reported results. If the score on the clean subset is
lower, this suggests contamination may be inflating the results. The results are summarized in Figure
C.2. Although potential contamination is often high (with a quarter of benchmarks scoring over

**Figure C.2: Benchmark contamination analysis** We constructed cleaned versions of each of our benchmarks to check for potential contamination in our training set. The x-axis is a conservative lower bound for how much of the dataset is known with high confidence to be clean, and the y-axis shows the difference in performance when evaluating only on the verified clean subset. Performance on most benchmarks changed negligibly, but some were flagged for further review. On inspection we find some evidence for contamination of the PIQA and Winograd results, and we mark the corresponding results with an asterisk. We find no evidence that other benchmarks are affected.

50%), in most cases performance changes only negligibly, and we see no evidence that contamination level and performance difference are correlated. We conclude that either our conservative method substantially overestimated contamination or that contamination has little effect on performance.

Below, we review in more detail the few specific cases where either (1) the model performs significantly worse on the cleaned version, or (2) potential contamination is very high, which makes measuring the performance difference difficult.

Our analysis flagged six groups of benchmarks for further investigation: Word Scrambling, Reading Comprehension (QuAC, SQuAD2, DROP), PIQA, Winograd, language modeling tasks (Wikitext tasks, 1BW), and German to English translation. Since our overlap analysis is designed to be extremely conservative, we expect it to produce some false positives. We summarize the results for each group of tasks below:

- **Reading Comprehension:** Our initial analysis flagged >90% of task examples from QuAC, SQuAD2, and DROP as potentially contaminated, so large that even measuring the differential on a clean subset was difficult. Upon manual inspection, however, we found that for every overlap we inspected, in all 3 datasets, the source text was present in our training data but the question/answer pairs were not, meaning the model gains only background information and cannot memorize the answer to a specific question.

- **German translation:** We found 25% of the examples in the WMT16 German-English test set were marked as potentially contaminated, with an associated total effect size of 1-2 BLEU. Upon inspection, none of the flagged examples contain paired sentences resembling NMT training data and collisions were monolingual matches mostly of snippets of events discussed in the news.

- **Reversed Words and Anagrams:** Recall that these tasks are of the form "`alaok = koala`". Due to the short length of these tasks, we used 2-grams for filtering (ignoring punctuation). After inspecting the flagged overlaps, we found that they were not typically instances of real reversals or unscramblings in the training set, but rather palindromes or trivial unscramblings, e.g "`kayak = kayak`". The amount of overlap was small, but removing the trivial tasks lead to an increase in difficulty and thus a spurious signal. Related to this, the symbol insertion task shows high overlap but no effect on performance – this is because that task involves removing non-letter characters from a word, and the overlap analysis itself ignores such characters, leading to many spurious matches.

- **PIQA:** The overlap analysis flagged 29% of examples as contaminated, and observed a 3 percentage point absolute decrease (4% relative decrease) in performance on the clean

subset. Though the test dataset was released after our training set was created and its labels are hidden, some of the web pages used by the crowdsourced dataset creators are contained in our training set. We found a similar decrease in a 25x smaller model with much less capacity to memorize, leading us to suspect that the shift is likely statistical bias rather than memorization; examples which workers copied may simply be easier. Unfortunately, we cannot rigorously prove this hypothesis. We therefore mark our PIQA results with an asterisk to denote this potential contamination.

- **Winograd:** The overlap analysis flagged 45% of examples, and found a 2.6% decrease in performance on the clean subset. Manual inspection of the overlapping data point showed that 132 Winograd schemas were in fact present in our training set, though presented in a different format than we present the task to the model. Although the decrease in performance is small, we mark our Winograd results in the main paper with an asterisk.

- **Language modeling:** We found the 4 Wikipedia language modeling benchmarks measured in GPT-2, plus the Children's Book Test dataset, to be almost entirely contained in our training data. Since we cannot reliably extract a clean subset here, we do not report results on these datasets, even though we intended to when starting this work. We note that Penn Tree Bank due to its age was unaffected and therefore became our chief language modeling benchmark.

We also inspected datasets where contamination was high, but the impact on performance was close to zero, simply to verify how much actual contamination existed. These appeared to often contain false positives. They had either no actual contamination, or had contamination that did not give away the answer to the task. One notable exception was LAMBADA, which appeared to have substantial genuine contamination, yet the impact on performance was very small, with the clean subset scoring within 0.5% of the full dataset. Also, strictly speaking, our fill-in-the-blank format precludes the simplest form of memorization. Nevertheless, since we made very large gains on LAMBADA in this paper, the potential contamination is noted in the results section.

An important limitation of our contamination analysis is that we cannot be sure that the clean subset is drawn from the same distribution as the original dataset. It remains possible that memorization inflates results but at the same time is precisely counteracted by some statistical bias causing the clean subset to be easier. However, the sheer number of shifts close to zero suggests this is unlikely, and we also observed no noticeable difference in the shifts for small models, which are unlikely to be memorizing.

**Intial training set filtering** We attempted to remove text occurring in benchmarks from training data by searching for $13-$gram overlaps between all test/development sets used in this work and our training data, and we removed the colliding $13-$gram as well as a 200 character window around it, splitting the original document into pieces. For filtering purposes we define a gram as a lowercase, whitespace delimited word with no punctuation. Pieces less than 200 characters long were discarded. Documents split into more than 10 pieces were considered contaminated and removed entirely. Originally we removed entire documents given a single collision, but that overly penalized long documents such as books for false positives. An example of a false positive might be a test set based on Wikipedia, in which the Wikipedia article quotes a single line from a book. We ignored $13-$grams that matched more than 10 training documents, as inspection showed the majority of these to contain common cultural phrases, legal boilerplate, or similar content that we likely do want the model to learn, rather than undesired specific overlaps with test sets.

Table C.1 shows the final mixture of datasets that we used in training. The CommonCrawl data was downloaded from 41 shards of monthly CommonCrawl covering 2016 to 2019, constituting 45TB of compressed plaintext before filtering and 570GB after filtering, roughly equivalent to 400 billion byte-pair-encoded tokens. Note that during training, datasets are not sampled in proportion to their size, but rather datasets we view as higher-quality are sampled more frequently, such that CommonCrawl and Books2 datasets are sampled less than once during training, but the other datasets are sampled 2-3 times. This essentially accepts a small amount of overfitting in exchange for higher quality training data.

**Overlap methodology** For our benchmark overlap analysis we used a variable number of words $N$ to check for overlap for each dataset, where $N$ is the 5th percentile example length in words, ignoring all punctuation, whitespace, and casing. Due to spurious collisions at lower values of $N$ we use a

| Dataset | Quantity (tokens) | Weight in training mix | Epochs elapsed when training for 300B tokens |
|---|---|---|---|
| Common Crawl (filtered) | 410 billion | 60% | 0.44 |
| WebText2 | 19 billion | 22% | 2.9 |
| Books1 | 12 billion | 8% | 1.9 |
| Books2 | 55 billion | 8% | 0.43 |
| Wikipedia | 3 billion | 3% | 3.4 |

**Table C.1: Datasets used to train GPT-3**. "Weight in training mix" refers to the fraction of examples during training that are drawn from a given dataset, which we intentionally do not make proportional to the size of the dataset. As a result, when we train for 300 billion tokens, some datasets are seen up to 3.4 times during training while other datasets are seen less than once.

minimum value of 8 on non-synthetic tasks. For performance reasons, we set a maximum value of 13 for all tasks. Values for $N$ and the amount of data marked as dirty are shown in Table C.2. Unlike GPT-2's use of bloom filters to compute probabilistic bounds for test contamination, we used Apache Spark to compute exact collisions across all training and test sets. We compute overlaps between test sets and our full training corpus, even though we only trained on 40% of our filtered Common Crawl documents per Table C.1.

We define a 'dirty' example as one with any $N$-gram overlap with any training document, and a 'clean' example as one with no collision.

Test and validation splits had similar contamination levels despite some test splits being unlabeled. Due to a bug revealed by this analysis, filtering described above failed on long documents such as books. Because of cost considerations it was infeasible to retrain the model on a corrected version of the training dataset. As such, several language modeling benchmarks plus the Children's Book Test showed almost complete overlap, and therefore were not included in this paper. Overlaps are shown in Table C.2

**Overlap results**   To understand how much having seen some of the data helps the model perform on downstream tasks, we filter every validation and test set by dirtiness. Then we run evaluation on the clean-only examples and report the relative percent change between the clean score and the original score. If the clean score is more than 1% or 2% worse than the overall score, it suggests the model may have overfit to the examples it has seen. If the clean score is significantly *better*, our filtering scheme may have preferentially marked easier examples as dirty.

This overlap metric tends to show a high rate of false positives for datasets that contain background information (but not answers) drawn from the web (such as SQuAD, which draws from Wikipedia) or examples less than 8 words long, which we ignored in our filtering process (except for wordscrambling tasks). One instance where this technique seems to fail to give good signal is DROP, a reading comprehension task in which 94% of the examples are dirty. The information required to answer the question is in a passage provided to the model, so having seen the passage during training but not the questions and answers does not meaningfully constitute cheating. We confirmed that every matching training document contained only the source passage, and none of the questions and answers in the dataset. The more likely explanation for the decrease in performance is that the 6% of examples that remain after filtering come from a slightly different distribution than the dirty examples.

Figure C.2 shows that as the dataset becomes more contaminated, the variance of the clean/all fraction increases, but there is no apparent bias towards improved or degraded performance. This suggests that GPT-3 is relatively insensitive to contamination.

Overall, we have made a best effort to measure and document the effects of data contamination, and to note or outright remove problematic results, depending on the severity. Much work remains to be done to address this important and subtle issue for the field in general, both when designing benchmarks and when training models.

| Name | Split | Metric | $N$ | Acc/F1/BLEU | Total Count | Dirty Acc/F1/BLEU | Dirty Count | Clean Acc/F1/BLEU | Clean Count | Clean Percentage | Relative Difference Clean vs All |
|---|---|---|---|---|---|---|---|---|---|---|---|
| Quac | dev | f1 | 13 | 44.3 | 7353 | 44.3 | 7315 | 54.1 | 38 | 1% | 20% |
| SQuADv2 | dev | f1 | 13 | 69.8 | 11873 | 69.9 | 11136 | 68.4 | 737 | 6% | -2% |
| DROP | dev | f1 | 13 | 36.5 | 9536 | 37.0 | 8898 | 29.5 | 638 | 7% | -21% |
| Symbol Insertion | dev | acc | 7 | 66.9 | 10000 | 66.8 | 8565 | 67.1 | 1435 | 14% | 0% |
| CoQa | dev | f1 | 13 | 86.0 | 7983 | 85.3 | 5107 | 87.1 | 2876 | 36% | 1% |
| ReCoRD | dev | acc | 13 | 89.5 | 10000 | 90.3 | 6110 | 88.2 | 3890 | 39% | -1% |
| Winograd | test | acc | 9 | 88.6 | 273 | 90.2 | 164 | 86.2 | 109 | 40% | -3% |
| BoolQ | dev | acc | 13 | 76.0 | 3270 | 75.8 | 1955 | 76.3 | 1315 | 40% | 0% |
| MultiRC | dev | acc | 13 | 74.2 | 953 | 73.4 | 558 | 75.3 | 395 | 41% | 1% |
| RACE-h | test | acc | 13 | 46.8 | 3498 | 47.0 | 1580 | 46.7 | 1918 | 55% | 0% |
| LAMBADA | test | acc | 13 | 86.4 | 5153 | 86.9 | 2209 | 86.0 | 2944 | 57% | 0% |
| LAMBADA (No Blanks) | test | acc | 13 | 77.8 | 5153 | 78.5 | 2209 | 77.2 | 2944 | 57% | -1% |
| WSC | dev | acc | 13 | 76.9 | 104 | 73.8 | 42 | 79.0 | 62 | 60% | 3% |
| PIQA | dev | acc | 8 | 82.3 | 1838 | 89.9 | 526 | 79.3 | 1312 | 71% | -4% |
| RACE-m | test | acc | 13 | 58.5 | 1436 | 53.0 | 366 | 60.4 | 1070 | 75% | 3% |
| De→En 16 | test | bleu-sb | 12 | 43.0 | 2999 | 47.4 | 739 | 40.8 | 2260 | 75% | -5% |
| En→De 16 | test | bleu-sb | 12 | 30.9 | 2999 | 32.6 | 739 | 29.9 | 2260 | 75% | -3% |
| En→Ro 16 | test | bleu-sb | 12 | 25.8 | 1999 | 24.9 | 423 | 26.1 | 1576 | 79% | 1% |
| Ro→En 16 | test | bleu-sb | 12 | 41.3 | 1999 | 40.4 | 423 | 41.6 | 1576 | 79% | 1% |
| WebQs | test | acc | 8 | 41.5 | 2032 | 41.6 | 428 | 41.5 | 1604 | 79% | 0% |
| ANLI R1 | test | acc | 13 | 36.8 | 1000 | 40.5 | 200 | 35.9 | 800 | 80% | -3% |
| ANLI R2 | test | acc | 13 | 34.0 | 1000 | 29.4 | 177 | 35.0 | 823 | 82% | 3% |
| TriviaQA | dev | acc | 10 | 71.2 | 7993 | 70.8 | 1390 | 71.3 | 6603 | 83% | 0% |
| ANLI R3 | test | acc | 13 | 40.2 | 1200 | 38.3 | 196 | 40.5 | 1004 | 84% | 1% |
| En→Fr 14 | test | bleu-sb | 13 | 39.9 | 3003 | 38.3 | 411 | 40.3 | 2592 | 86% | 1% |
| Fr→En 14 | test | bleu-sb | 13 | 41.4 | 3003 | 40.9 | 411 | 41.4 | 2592 | 86% | 0% |
| WiC | dev | acc | 13 | 51.4 | 638 | 53.1 | 49 | 51.3 | 589 | 92% | 0% |
| RTE | dev | acc | 13 | 71.5 | 277 | 71.4 | 21 | 71.5 | 256 | 92% | 0% |
| CB | dev | acc | 13 | 80.4 | 56 | 100.0 | 4 | 78.8 | 52 | 93% | -2% |
| Anagrams 2 | dev | acc | 2 | 40.2 | 10000 | 76.2 | 705 | 37.4 | 9295 | 93% | -7% |
| Reversed Words | dev | acc | 2 | 0.4 | 10000 | 1.5 | 660 | 0.3 | 9340 | 93% | -26% |
| OpenBookQA | test | acc | 8 | 65.4 | 500 | 58.1 | 31 | 65.9 | 469 | 94% | 1% |
| ARC (Easy) | test | acc | 11 | 70.1 | 2268 | 77.5 | 89 | 69.8 | 2179 | 96% | 0% |
| Anagrams 1 | dev | acc | 2 | 15.0 | 10000 | 49.8 | 327 | 13.8 | 9673 | 97% | -8% |
| COPA | dev | acc | 9 | 93.0 | 100 | 100.0 | 3 | 92.8 | 97 | 97% | 0% |
| ARC (Challenge) | test | acc | 12 | 51.6 | 1144 | 45.2 | 31 | 51.8 | 1113 | 97% | 0% |
| HellaSwag | dev | acc | 13 | 79.3 | 10042 | 86.2 | 152 | 79.2 | 9890 | 98% | 0% |
| NQs | test | acc | 11 | 29.9 | 3610 | 32.7 | 52 | 29.8 | 3558 | 99% | 0% |
| Cycled Letters | dev | acc | 2 | 38.6 | 10000 | 20.5 | 73 | 38.7 | 9927 | 99% | 0% |
| SAT Analogies | dev | acc | 9 | 65.8 | 374 | 100.0 | 2 | 65.6 | 372 | 99% | 0% |
| StoryCloze | test | acc | 13 | 87.7 | 1871 | 100.0 | 2 | 87.6 | 1869 | 100% | 0% |
| Winogrande | dev | acc | 13 | 77.7 | 1267 | - | 0 | 77.7 | 1267 | 100% | 0% |

**Table C.2:** Overlap statistics for all datasets sorted from dirtiest to cleanest. We consider a dataset example dirty if it has a single $N$-gram collision with any document in our training corpus. "Relative Difference Clean vs All" shows the percent change in performance between only the clean examples vs all the examples in the benchmark. "Count" shows the number of examples. "Clean percentage" is the percent of examples that are clean vs total. For "Acc/F1/BLEU" we use the metric specified in "Metric". These scores come from evaluations with a different seed for the random examples used for in-context learning, and will therefore differ slightly from the scores elsewhere in the paper.

## D   Total Compute Used to Train Language Models

This appendix contains the calculations that were used to derive the approximate compute used to train the language models. As a simplifying assumption, we ignore the attention operation, as it typically uses less than 10% of the total compute for the models we are analyzing.

Calculations can be seen in Table D.1 and are explained within the table caption.

| Model | Total train compute (PF-days) | Total train compute (flops) | Params (M) | Training tokens (billions) | Flops per param per token | Mult for bwd pass | Fwd-pass flops per active param per token | Frac of params active for each token |
|---|---|---|---|---|---|---|---|---|
| T5-Small | 2.08E+00 | 1.80E+20 | 60 | 1,000 | 3 | 3 | 1 | 0.5 |
| T5-Base | 7.64E+00 | 6.60E+20 | 220 | 1,000 | 3 | 3 | 1 | 0.5 |
| T5-Large | 2.67E+01 | 2.31E+21 | 770 | 1,000 | 3 | 3 | 1 | 0.5 |
| T5-3B | 1.04E+02 | 9.00E+21 | 3,000 | 1,000 | 3 | 3 | 1 | 0.5 |
| T5-11B | 3.82E+02 | 3.30E+22 | 11,000 | 1,000 | 3 | 3 | 1 | 0.5 |
| BERT-Base | 1.89E+00 | 1.64E+20 | 109 | 250 | 6 | 3 | 2 | 1.0 |
| BERT-Large | 6.16E+00 | 5.33E+20 | 355 | 250 | 6 | 3 | 2 | 1.0 |
| RoBERTa-Base | 1.74E+01 | 1.50E+21 | 125 | 2,000 | 6 | 3 | 2 | 1.0 |
| RoBERTa-Large | 4.93E+01 | 4.26E+21 | 355 | 2,000 | 6 | 3 | 2 | 1.0 |
| GPT-3 Small | 2.60E+00 | 2.25E+20 | 125 | 300 | 6 | 3 | 2 | 1.0 |
| GPT-3 Medium | 7.42E+00 | 6.41E+20 | 356 | 300 | 6 | 3 | 2 | 1.0 |
| GPT-3 Large | 1.58E+01 | 1.37E+21 | 760 | 300 | 6 | 3 | 2 | 1.0 |
| GPT-3 XL | 2.75E+01 | 2.38E+21 | 1,320 | 300 | 6 | 3 | 2 | 1.0 |
| GPT-3 2.7B | 5.52E+01 | 4.77E+21 | 2,650 | 300 | 6 | 3 | 2 | 1.0 |
| GPT-3 6.7B | 1.39E+02 | 1.20E+22 | 6,660 | 300 | 6 | 3 | 2 | 1.0 |
| GPT-3 13B | 2.68E+02 | 2.31E+22 | 12,850 | 300 | 6 | 3 | 2 | 1.0 |
| GPT-3 175B | 3.64E+03 | 3.14E+23 | 174,600 | 300 | 6 | 3 | 2 | 1.0 |

**Table D.1:** Starting from the right hand side and moving left, we begin with the number of training tokens that each model was trained with. Next we note that since T5 uses an encoder-decoder model, only half of the parameters are active for each token during a forward or backwards pass. We then note that each token is involved in a single addition and a single multiply for each active parameter in the forward pass (ignoring attention). Then we add a multiplier of 3x to account for the backwards pass (as computing both $\frac{\partial params}{\partial loss}$ and $\frac{\partial acts}{\partial loss}$ use a similar amount of compute as the forwards pass. Combining the previous two numbers, we get the total flops per parameter per token. We multiply this value by the total training tokens and the total parameters to yield the number of total flops used during training. We report both flops and petaflop/s-day (each of which are 2.88e+7 flops).

## E   Human Quality Assessment of Synthetic News Articles

This appendix contains details on the experiments measuring human ability to distinguish GPT-3-generated synthetic news articles from real news articles. We first describe the experiments on the $\sim 200$ word news articles, and then describe the preliminary investigation of $\sim 500$ word news articles generated by GPT-3.

*Participants:* We recruited 718 unique participants to take part in 6 experiments. 97 participants were excluded for failing an internet check question, leaving a total of 621 participants: 343 male, 271 female, and 7 other. Mean participant age was $\sim 38$ years old. All participants were recruited through Positly, which maintains a whitelist of high-performing workers from Mechanical Turk. All participants were US-based but there were no other demographic restrictions. Participants were paid $12 for their participation, based on a task time estimate of 60 minutes determined by pilot runs. In order to ensure that the sample of participants for each experiment quiz was unique, participants were not allowed to take part in an experiment more than once.

*Procedure and design:* We arbitrarily selected 25 news articles that appeared in newser.com in early 2020. We used the article titles and subtitles to produce outputs from the 125M, 350M, 760M, 1.3B, 2.7B, 6.7B, 13.0B, and 200B (GPT-3) parameter language models. Five outputs per question were

**Figure E.1:** Participants spend more time trying to identify whether each news article is machine generated as model size increases. Duration on the control model is indicated with the dashed line. Line of best fit is a linear model on a log scale with 95% confidence intervals.

generated by each model and the generation with a word count closest to that of the human written article was selected automatically. This was to minimize the effect that completion length might have on participants' judgments. The same output procedure for each model with the exception of the removal of the intentionally bad control model, as described in the main text.

In each experiment, half of the participants were randomly assigned to quiz A and half were randomly assigned to quiz B. Each quiz consisted of 25 articles: half (12-13) were human written and half (12-13) were model generated: the articles with human written completions in quiz A had model generated completions in quiz B and vice versa. The order of quiz question was shuffled for each participant. Participants could leave comments and were asked to indicate if they had seen the articles before. Participants were instructed not to look up the articles or their content during the quiz and at the end of the quiz were asked if they had looked anything up during the quiz.

*Statistical Tests:* To compare means on the different runs, we performed a two-sample t-test for independent groups for each model against the control. This was implemented in Python using the `scipy.stats.ttest_ind` function. When plotting a regression line in the graph of average participant accuracy vs model size, we fit a power law of the form $ax^{-b}$. The 95% confidence intervals were estimated from the t-distribution of the sample mean.

*Duration statistics*: In the main text, we discussed the finding that the ability of human participants to distinguish model and human generated news articles decreases as our models become larger. We have also found that the average time spent for a given set of questions increases as the model size increases, as shown in Figure E.1. Lower accuracy scores despite increased time investment from participants supports the finding that larger models generate harder-to-distinguish news articles.

*Preliminary investigation of* $\sim 500$ *word articles:* We recruited 160 unique US-based participants to take part in 2 experiments through Positly (details are given in Table E.1). We randomly selected 12 Reuters world news articles from late 2019 and created a context for GPT-3 175B that consisted of a single Reuters article not in this set of 12. We then used the article titles and Reuters locations to generate completions from GPT-3 175B and the 160M control model from the previous experiments. These were used to create two 12-question quizzes per model, each consisting of half human written and half model generated articles. Comprehension questions were added and articles were shown to participants in 3 stages at 30 second intervals to encourage closer reading. Participants were paid $12 for this task. Model generation selection methods, exclusion criteria, and statistical tests mirror those of the previous experiments.

| Model | Participants Recruited | Participants Excluded | Genders (m:f:other) | Mean Age | Average Word Count (human:model) |
|---|---|---|---|---|---|
| Control | 79 | 17 | 32:37:0 | 39 | 569:464 |
| GPT-3 175B | 81 | 19 | 32:30:0 | 40 | 569:498 |

**Table E.1:** Participant details and article lengths for the experiments investigating human detection of $\sim 500$ word model generated news articles. Participants were excluded due to internet check fails.

# F   Additional Samples from GPT-3

GPT-3 adapts well to many tasks other than the ones explored in the main body of the paper. As an example, in Figure F.1, we show four uncurated samples from a prompt suggesting that the model write a poem, with a given title, in the style of Wallace Stevens. We first experimented with a few prompts, then generated four samples with no additional editing or selection (sampling at temperature 1 using nucleus sampling [HBFC19] with $P = 0.9$). Completions were truncated when the model began to write a new title and author heading, or broke into prose commentary.

```
Context →                                    The City
                                             BY C. P. CAVAFY
                                             TRANSLATED BY EDMUND KEELEY
                                             [Poem text omitted]

                                             SOME TREES
                                             John Ashbery
                                             [Poem text omitted]

                                             Shadows on the Way
                                             Wallace Stevens
```

```
-------- Generated Poem 1 --------           -------- Generated Poem 3 --------

I must have shadows on the way               The sun was all we had.  Now, in the shade
If I am to walk I must have                  All is changed.  The mind must dwell on those
Each step taken slowly and alone             White fields, that to its eyes were always old;
To have it ready made                        Those ancient gleams, convoluting
                                             The way they lay among the huge roots,
And I must think in lines of grey            The great dark tomes of reverie,
To have dim thoughts to be my guide          The plumed labyrinth of the sea.
Must look on blue and green
And never let my eye forget                  When our own breath gonne faint and fade
That color is my friend                      Up the long stairway of the dead
And purple must surround me too              We shall remember well
                                             The blindness of the light, the whiteness
The yellow of the sun is no more             Of that white land.  The footsteps, and
Intrusive than the bluish snow               The grasses in that light, and in the shade
That falls on all of us.  I must have        The menacing life.
Grey thoughts and blue thoughts walk with me
If I am to go away at all.                   We played, too, once, in the warmest rooms.
                                             To one content, there is one content
-------- Generated Poem 2 --------           For all mankind.  The forest glades
                                             Are all the more for being in fable,
There is a way in the world.                 And wood and town in legend shadowed-
That way a man may live in a small town      Once more to return.  The one content
His lifetime and see no man come,            Has in it every happiness, more brave
Never the stir of men about a market,        For being purposeless; and so
Never the bobbing of covered carts,          Those island gardens, deep in the mind,
Never the creak passed on the stair          Make of remoteness familiarity
Closing some other life; an old man          And of the unspoken name a friend.
Can prosper in that town, can prosper
When his store is seen to close, his wife    -------- Generated Poem 4 --------
Fades in his house, and the world shouts in vain
That no one comes to hear him                Nobody will come to this place.  It is a road that leads nowhere.
Where his small bell tolls.                  The solitude is deep.  The mountains are high.
In that small town                           But they are desolate, and they turn the traveler's face
Everything is known.                         Towards the North.  All the sounds of the world are far away.
The ways of the young men are known,         When the wind rises above the trees,
And the ways of the young women              The boughs bow to the ground.
As they walk to the Spring;                  Even the birds that inhabit the tangle of weeds
The rumors are all known,                    That is the roadside cover, are silent.  One listens,
The quarrels, the hurt, the envy             But hears no roar of the forest.  One is alone.
Of the men and the women.                    One will be taken.
Till a day comes                             One will be taken.
That way out of the world                    There is no utterance, there is no conversation,
That leads to a wood.                         But one is uneasy all the same....
Young, the young man                         There is a thin blue mist,
Walks, a lover now,                          A darkness rising like smoke,
Knows the streets as they are and is of them. And within that darkness
                                             A possession of the heart.
Comes to the wood.                           One will be taken....  It was here, and it will be here again-
                                             Here, under this sky empty and full of light.
There, in the wood, among trees,
He sees shadows on the way, hears voices,
hears the wind and the rustling of leaves;
Through an open glade
He sees a shape and the shape hears:
It waits as he waits,
Waits as the shadows wait,
As the voices wait;
Shadows on the way, voices in the wind.
```

**Figure F.1:** Four uncurated completions from a context suggesting the model compose a poem in the style of Wallace Stevens with the title 'Shadows on the Way'.

# G   Details of Task Phrasing and Specifications

The following figures illustrate the formatting and phrasing of all the tasks included in the paper. All data comes from the ground truth datasets in this section, and no samples from GPT-3 are included here.

```
       Context →   Article:
                   Informal conversation is an important part of any business
                   relationship.Before you start a discussion,however,make
                   sure you understand which topics are suitable and which are
                   considered taboo in a particular culture.  Latin Americans
                   enjoy sharing information about their local history, art
                   and customs.You may expect questions about your family,and
                   be sure to show pictures of your children.You may feel free
                   to ask similar questions of your Latin American friends.The
                   French think of conversation as an art form,and they enjoy
                   the value of lively discussions as well as disagreements.
                   For them,arguments can be interesting and they can cover
                   pretty much or any topic ---- as long as they occur in are
                   respectful and intelligent manner.
                   In the United States,business people like to discuss
                   a wide range of topics,including opinions about
                   work,family,hobbies,and politics.  In Japan,China,and
                   Korea,however,people are much more private.They do not
                   share much about their thoughts,feelings,or emotions because
                   they feel that doing so might take away from the harmonious
                   business relationship they're trying to build.Middle
                   Easterners are also private about their personal lives and
                   family matters.It is considered rude,for example,to ask a
                   businessman from Saudi Arabia about his wife or children.
                   As a general rule,it's best not to talk about politics
                   or religion with your business friends.This can get you
                   into trouble,even in the United States,where people hold
                   different religious views.In addition,discussing one's
                   salary is usually considered unsuitable.Sports is typically
                   a friendly subject in most parts of the world,although be
                   careful not to criticize national sport.Instead,be friendly
                   and praise your host's team.

                   Q: What shouldn't you do when talking about sports with
                   colleagues from another country?

                   A: Criticizing the sports of your colleagues' country.

                   Q: Which is typically a friendly topic in most places
                   according to the author?

                   A: Sports.

                   Q: Why are people from Asia more private in their
                   conversation with others?

                   A: They don't want to have their good relationship with
                   others harmed by informal conversation.

                   Q: The author considers politics and religion _ .

                   A:
  Correct Answer →   taboo
Incorrect Answer →   cheerful topics
Incorrect Answer →   rude topics
Incorrect Answer →   topics that can never be talked about
```

**Figure G.1:** Formatted dataset example for RACE-h.  When predicting, we normalize by the unconditional probability of each answer.

| | |
|---|---|
| Context → | `anli 2:  anli 2:  The Gold Coast Hotel & Casino is a hotel`<br>`and casino located in Paradise, Nevada.  This locals' casino`<br>`is owned and operated by Boyd Gaming.  The Gold Coast is`<br>`located one mile (∼    1.6km) west of the Las Vegas Strip on`<br>`West Flamingo Road.  It is located across the street from`<br>`the Palms Casino Resort and the Rio All Suite Hotel and`<br>`Casino.`<br>`Question:  The Gold Coast is a budget-friendly casino.`<br>`True, False, or Neither?` |
| Correct Answer → | `Neither` |
| Incorrect Answer → | `True` |
| Incorrect Answer → | `False` |

**Figure G.2:** Formatted dataset example for ANLI R2

```
        Context →   Article:
                    Mrs.  Smith is an unusual teacher.  Once she told each
                    student to bring along a few potatoes in plastic bag.  On
                    each potato the students had to write a name of a person
                    that they hated And the next day, every child brought some
                    potatoes.  Some had two potatoes;some three;some up to five.
                    Mrs.  Smith then told the children to carry the bags
                    everywhere they went, even to the toilet, for two weeks.
                    As day after day passed, the children started to complain
                    about the awful smell of the rotten potatoes.
                    Those children who brought five potatoes began to feel the
                    weight trouble of the bags.  After two weeks, the children
                    were happy to hear that the game was finally ended.  Mrs.
                    Smith asked,"How did you feel while carrying the potatoes
                    for two weeks?" The children started complaining about the
                    trouble loudly.
                    Then Mrs.  Smith told them why she asked them to play the
                    game.  She said,"This is exactly the situation when you
                    carry your hatred for somebody inside your heart.  The
                    terrible smell of the hatred will pollute your heart and you
                    will carry something unnecessary with you all the time.  If
                    you cannot stand the smell of the rotten potatoes for just
                    two weeks, can you imagine how heavy it would be to have the
                    hatred in your heart for your lifetime?  So throw away any
                    hatred from your heart, and you'll be really happy."

                    Q: Which of the following is True according to the passage?

                    A: If a kid hated four people,he or she had to carry four
                    potatoes.

                    Q: We can learn from the passage that we should _ .

                    A: throw away the hatred inside

                    Q: The children complained about _ besides the weight
                    trouble.

                    A: the smell

                    Q: Mrs.Smith asked her students to write _ on the potatoes.

                    A:
 Correct Answer →   names
Incorrect Answer →  numbers
Incorrect Answer →  time
Incorrect Answer →  places
```

**Figure G.3:** Formatted dataset example for RACE-m.  When predicting, we normalize by the unconditional probability of each answer.

```
        Context →   How to apply sealant to wood.
 Correct Answer →   Using a brush, brush on sealant onto wood until it is fully
                    saturated with the sealant.
Incorrect Answer →  Using a brush, drip on sealant onto wood until it is fully
                    saturated with the sealant.
```

**Figure G.4:** Formatted dataset example for PIQA

| | |
|---|---|
| Context → | My body cast a shadow over the grass because |
| Correct Answer → | the sun was rising. |
| Incorrect Answer → | the grass was cut. |

**Figure G.5:** Formatted dataset example for COPA

| | |
|---|---|
| Context → | (CNN) Yuval Rabin, whose father, Yitzhak Rabin, was assassinated while serving as Prime Minister of Israel, criticized Donald Trump for appealing to "Second Amendment people" in a speech and warned that the words that politicians use can incite violence and undermine democracy. "Trump's words are an incitement to the type of political violence that touched me personally," Rabin wrote in USAToday.  He said that Trump's appeal to "Second Amendment people" to stop Hillary Clinton -- comments that were criticized as a call for violence against Clinton, something Trump denied -- "were a new level of ugliness in an ugly campaign season."<br><br>- The son of a former Israeli Prime Minister who was assassinated wrote an op ed about the consequence of violent political rhetoric.<br>- Warns of "parallels" between Israel of the 1990s and the U.S. today. |
| Correct Answer → | - Referencing his father, who was shot and killed by an extremist amid political tension in Israel in 1995, Rabin condemned Donald Trump's aggressive rhetoric. |
| Correct Answer → | - Referencing his father, who was shot and killed by an extremist amid political tension in Israel in 1995, Rabin condemned Trump's aggressive rhetoric. |
| Incorrect Answer → | - Referencing his father, who was shot and killed by an extremist amid political tension in Israel in 1995, Rabin condemned Hillary Clinton's aggressive rhetoric. |
| Incorrect Answer → | - Referencing his father, who was shot and killed by an extremist amid political tension in Israel in 1995, Rabin condemned U.S.'s aggressive rhetoric. |
| Incorrect Answer → | - Referencing his father, who was shot and killed by an extremist amid political tension in Israel in 1995, Rabin condemned Yitzhak Rabin's aggressive rhetoric. |

**Figure G.6:** Formatted dataset example for ReCoRD. We consider the context above to be a single "problem" because this is how the task is presented in the ReCoRD dataset and scored in the ReCoRD evaluation script.

| | |
|---|---|
| Context → | anli 1:  anli 1:  Fulton James MacGregor MSP is a Scottish politician who is a Scottish National Party (SNP) Member of Scottish Parliament for the constituency of Coatbridge and Chryston.  MacGregor is currently Parliamentary Liaison Officer to Shona Robison, Cabinet Secretary for Health & Sport.  He also serves on the Justice and Education & Skills committees in the Scottish Parliament.<br>Question:  Fulton James MacGregor is a Scottish politician who is a Liaison officer to Shona Robison who he swears is his best friend.  True, False, or Neither? |
| Correct Answer → | Neither |
| Incorrect Answer → | True |
| Incorrect Answer → | False |

**Figure G.7:** Formatted dataset example for ANLI R1

| | |
|---:|:---|
| Context → | Organisms require energy in order to do what? |

| | |
|---:|:---|
| Correct Answer → | mature and develop. |
| Incorrect Answer → | rest soundly. |
| Incorrect Answer → | absorb light. |
| Incorrect Answer → | take in nutrients. |

**Figure G.8:** Formatted dataset example for OpenBookQA. When predicting, we normalize by the unconditional probability of each answer.

| | |
|---:|:---|
| Context → | Making a cake:  Several cake pops are shown on a display.  A woman and girl are shown making the cake pops in a kitchen.  They |

| | |
|---:|:---|
| Correct Answer → | bake them, then frost and decorate. |
| Incorrect Answer → | taste them as they place them on plates. |
| Incorrect Answer → | put the frosting on the cake as they pan it. |
| Incorrect Answer → | come out and begin decorating the cake as well. |

**Figure G.9:** Formatted dataset example for HellaSwag

| | |
|---:|:---|
| Context → | anli 3:  anli 3:  We shut the loophole which has American workers actually subsidizing the loss of their own job.  They just passed an expansion of that loophole in the last few days:  $43 billion of giveaways, including favors to the oil and gas industry and the people importing ceiling fans from China.  Question:  The loophole is now gone True, False, or Neither? |

| | |
|---:|:---|
| Correct Answer → | False |
| Incorrect Answer → | True |
| Incorrect Answer → | Neither |

**Figure G.10:** Formatted dataset example for ANLI R3

| | |
|---:|:---|
| Context → | Question:  George wants to warm his hands quickly by rubbing them.  Which skin surface will produce the most heat?  Answer: |

| | |
|---:|:---|
| Correct Answer → | dry palms |
| Incorrect Answer → | wet palms |
| Incorrect Answer → | palms covered with oil |
| Incorrect Answer → | palms covered with lotion |

**Figure G.11:** Formatted dataset example for ARC (Challenge). When predicting, we normalize by the unconditional probability of each answer.

| | |
|---:|:---|
| Context → | lull is to trust as |

| | |
|---:|:---|
| Correct Answer → | cajole is to compliance |
| Incorrect Answer → | balk is to fortitude |
| Incorrect Answer → | betray is to loyalty |
| Incorrect Answer → | hinder is to destination |
| Incorrect Answer → | soothe is to passion |

**Figure G.12:** Formatted dataset example for SAT Analogies

| | |
|---|---|
| Correct Context → | Grace was happy to trade me her sweater for my jacket. She thinks the sweater |
| Incorrect Context → | Grace was happy to trade me her sweater for my jacket. She thinks the jacket |
| Target Completion → | looks dowdy on her. |

**Figure G.13:** Formatted dataset example for Winograd. The 'partial' evaluation method we use compares the probability of the completion given a correct and incorrect context.

| | |
|---|---|
| Correct Context → | Johnny likes fruits more than vegetables in his new keto diet because the fruits |
| Incorrect Context → | Johnny likes fruits more than vegetables in his new keto diet because the vegetables |
| Target Completion → | are saccharine. |

**Figure G.14:** Formatted dataset example for Winogrande. The 'partial' evaluation method we use compares the probability of the completion given a correct and incorrect context.

```
        Context →    READING COMPREHENSION ANSWER KEY
                     While this process moved along, diplomacy continued
                     its rounds.  Direct pressure on the Taliban had proved
                     unsuccessful.  As one NSC staff note put it, "Under the
                     Taliban, Afghanistan is not so much a state sponsor of
                     terrorism as it is a state sponsored by terrorists." In
                     early 2000, the United States began a high-level effort to
                     persuade Pakistan to use its influence over the Taliban.  In
                     January 2000, Assistant Secretary of State Karl Inderfurth
                     and the State Department's counterterrorism coordinator,
                     Michael Sheehan, met with General Musharraf in Islamabad,
                     dangling before him the possibility of a presidential visit
                     in March as a reward for Pakistani cooperation.  Such a
                     visit was coveted by Musharraf, partly as a sign of his
                     government's legitimacy.  He told the two envoys that he
                     would meet with Mullah Omar and press him on Bin Laden.
                     They left, however, reporting to Washington that Pakistan
                     was unlikely in fact to do anything," given what it sees as
                     the benefits of Taliban control of Afghanistan." President
                     Clinton was scheduled to travel to India.  The State
                     Department felt that he should not visit India without also
                     visiting Pakistan.  The Secret Service and the CIA, however,
                     warned in the strongest terms that visiting Pakistan would
                     risk the President's life.  Counterterrorism officials
                     also argued that Pakistan had not done enough to merit
                     a presidential visit.  But President Clinton insisted on
                     including Pakistan in the itinerary for his trip to South
                     Asia.  His one-day stopover on March 25, 2000, was the
                     first time a U.S. president had been there since 1969.  At
                     his meeting with Musharraf and others, President Clinton
                     concentrated on tensions between Pakistan and India and
                     the dangers of nuclear proliferation, but also discussed
                     Bin Laden.  President Clinton told us that when he pulled
                     Musharraf aside for a brief, one-on-one meeting, he pleaded
                     with the general for help regarding Bin Laden." I offered
                     him the moon when I went to see him, in terms of better
                     relations with the United States, if he'd help us get Bin
                     Laden and deal with another issue or two." The U.S. effort
                     continued.

                     Who did The State Department feel should visit both India
                     and Pakistan?

   Correct Answer →  - [False] Bin Laden
 Incorrect Answer →  - [True] Bin Laden
```

**Figure G.15:** Formatted dataset example for MultiRC. There are three levels within MultiRC: (1) the passage, (2) the questions, and (3) the answers. During evaluation, accuracy is determined at the per-question level, with a question being considered correct if and only if all the answers within the question are labeled correctly. For this reason, we use $K$ to refer to the number of **questions** shown within the context.

```
        Context →    Question:  Which factor will most likely cause a person to
                     develop a fever?
                     Answer:

   Correct Answer →  a bacterial population in the bloodstream
 Incorrect Answer →  a leg muscle relaxing after exercise
 Incorrect Answer →  several viral particles on the skin
 Incorrect Answer →  carbohydrates being digested in the stomach
```

**Figure G.16:** Formatted dataset example for ARC (Easy). When predicting, we normalize by the unconditional probability of each answer.

|  |  |
|---|---|
| Context → | Bob went to the gas station to fill up his car. His tank was completely empty and so was his wallet. The cashier offered to pay for his gas if he came back later to pay. Bob felt grateful as he drove home. |
| Correct Answer → | Bob believed that there were good people in the world. |
| Incorrect Answer → | Bob contemplated how unfriendly the world was. |

**Figure G.17:** Formatted dataset example for StoryCloze

|  |  |
|---|---|
| Context → | Helsinki is the capital and largest city of Finland. It is in the region of Uusimaa, in southern Finland, on the shore of the Gulf of Finland. Helsinki has a population of , an urban population of , and a metropolitan population of over 1.4 million, making it the most populous municipality and urban area in Finland. Helsinki is some north of Tallinn, Estonia, east of Stockholm, Sweden, and west of Saint Petersburg, Russia. Helsinki has close historical connections with these three cities.

The Helsinki metropolitan area includes the urban core of Helsinki, Espoo, Vantaa, Kauniainen, and surrounding commuter towns. It is the world's northernmost metro area of over one million people, and the city is the northernmost capital of an EU member state. The Helsinki metropolitan area is the third largest metropolitan area in the Nordic countries after Stockholm and Copenhagen, and the City of Helsinki is the third largest after Stockholm and Oslo. Helsinki is Finland's major political, educational, financial, cultural, and research center as well as one of northern Europe's major cities. Approximately 75% of foreign companies that operate in Finland have settled in the Helsinki region. The nearby municipality of Vantaa is the location of Helsinki Airport, with frequent service to various destinations in Europe and Asia.

Q: what is the most populous municipality in Finland?

A: Helsinki

Q: how many people live there?

A: 1.4 million in the metropolitan area

Q: what percent of the foreign companies that operate in Finland are in Helsinki?

A: 75%

Q: what towns are a part of the metropolitan area?

A: |
| Target Completion → | Helsinki, Espoo, Vantaa, Kauniainen, and surrounding commuter towns |

**Figure G.18:** Formatted dataset example for CoQA

| | |
|---|---|
| Context → | Please unscramble the letters into a word, and write that word:<br>asinoc = |
| Target Completion → | casino |

**Figure G.19:** Formatted dataset example for Cycled Letters

| | |
|---|---|
| Context → | Passage: Saint Jean de Brébeuf was a French Jesuit missionary who travelled to New France in 1625. There he worked primarily with the Huron for the rest of his life, except for a few years in France from 1629 to 1633. He learned their language and culture, writing extensively about each to aid other missionaries. In 1649, Brébeuf and another missionary were captured when an Iroquois raid took over a Huron village . Together with Huron captives, the missionaries were ritually tortured and killed on March 16, 1649. Brébeuf was beatified in 1925 and among eight Jesuit missionaries canonized as saints in the Roman Catholic Church in 1930.<br>Question: How many years did Saint Jean de Brébeuf stay in New France before he went back to France for a few years?<br>Answer: |
| Target Completion → | 4 |

**Figure G.20:** Formatted dataset example for DROP

| | |
|---|---|
| Context → | Fill in blank:<br><br>She held the torch in front of her.<br><br>She caught her breath.<br><br>"Chris? There's a step."<br><br>"What?"<br><br>"A step. Cut in the rock. About fifty feet ahead." She moved faster. They both moved faster. "In fact," she said, raising the torch higher, "there's more than a ____.<br>-> |
| Target Completion → | step |

**Figure G.21:** Formatted dataset example for LAMBADA

| | |
|---|---|
| Context → | Please unscramble the letters into a word, and write that word:<br>skicts = |
| Target Completion → | sticks |

**Figure G.22:** Formatted dataset example for Anagrams 1 (A1)

| | |
|---|---|
| Context → | Please unscramble the letters into a word, and write that word:<br>volwskagen = |
| Target Completion → | volkswagen |

**Figure G.23:** Formatted dataset example for Anagrams 2

| | |
|---|---|
| Context → | Q: Who played tess on touched by an angel?<br><br>A: |
| Target Completion → | Delloreese Patricia Early (July 6, 1931 { November 19, 2017), known professionally as Della Reese |

**Figure G.24:** Formatted dataset example for Natural Questions

| | |
|---|---|
| Context → | TITLE: William Perry (American football) - Professional career<br>PARAGRAPH: In 1985, he was selected in the first round of the 1985 NFL Draft by the Chicago Bears; he had been hand-picked by coach Mike Ditka. However, defensive coordinator Buddy Ryan, who had a highly acrimonious relationship with Ditka, called Perry a "wasted draft-pick". Perry soon became a pawn in the political power struggle between Ditka and Ryan. Perry's "Refrigerator" nickname followed him into the NFL and he quickly became a favorite of the Chicago Bears fans. Teammates called him "Biscuit," as in "one biscuit shy of 350 pounds." While Ryan refused to play Perry, Ditka decided to use Perry as a fullback when the team was near the opponents' goal line or in fourth and short situations, either as a ball carrier or a lead blocker for star running back Walter Payton. Ditka stated the inspiration for using Perry as a fullback came to him during five-yard sprint exercises. During his rookie season, Perry rushed for two touchdowns and caught a pass for one. Perry even had the opportunity to run the ball during Super Bowl XX, as a nod to his popularity and contributions to the team's success. The first time he got the ball, he was tackled for a one-yard loss while attempting to throw his first NFL pass on a halfback option play. The second time he got the ball, he scored a touchdown (running over Patriots linebacker Larry McGrew in the process). About halfway through his rookie season, Ryan finally began to play Perry, who soon proved that he was a capable defensive lineman. His Super Bowl ring size is the largest of any professional football player in the history of the event. His ring size is 25, while the ring size for the average adult male is between 10 and 12. Perry went on to play for ten years in the NFL, retiring after the 1994 season. In his ten years as a pro, he regularly struggled with his weight, which hampered his performance at times. He played in 138 games, recording 29.5 sacks and five fumble recoveries, which he returned for a total of 71 yards. In his offensive career he ran five yards for two touchdowns, and had one reception for another touchdown. Perry later attempted a comeback, playing an unremarkable 1996 season with the London Monarchs of the World League of American Football (later NFL Europa).<br><br>Q: what team did he play for?<br><br>A: |
| Target Completion → | the Chicago Bears |

**Figure G.25:** Formatted dataset example for QuAC

| Context $\rightarrow$ | Please unscramble the letters into a word, and write that word:<br>r e!c.i p r o.c a/l = |
|---|---|
| Target Completion $\rightarrow$ | reciprocal |

**Figure G.26:** Formatted dataset example for Symbol Insertion

| Context $\rightarrow$ | Please unscramble the letters into a word, and write that word:<br>taefed = |
|---|---|
| Target Completion $\rightarrow$ | defeat |

**Figure G.27:** Formatted dataset example for Reversed Words

| Context $\rightarrow$ | Title:  The_Blitz<br><br>Background:  From the German point of view, March 1941 saw an improvement.  The Luftwaffe flew 4,000 sorties that month, including 12 major and three heavy attacks.  The electronic war intensified but the Luftwaffe flew major inland missions only on moonlit nights.  Ports were easier to find and made better targets.  To confuse the British, radio silence was observed until the bombs fell.  X- and Y-Gerät beams were placed over false targets and switched only at the last minute.  Rapid frequency changes were introduced for X-Gerät, whose wider band of frequencies and greater tactical flexibility ensured it remained effective at a time when British selective jamming was degrading the effectiveness of Y-Gerät.<br><br>Q: How many sorties were flown in March 1941?<br><br>A: 4,000<br><br>Q: When did the Luftwaffe fly inland missions?<br><br>A: |
|---|---|
| Target Completion $\rightarrow$ | only on moonlit nights |

**Figure G.28:** Formatted dataset example for SQuADv2

| | |
|---|---|
| Context → | Normal force -- In a simple case such as an object resting upon a table, the normal force on the object is equal but in opposite direction to the gravitational force applied on the object (or the weight of the object), that is, N = m g (\displaystyle N=mg), where m is mass, and g is the gravitational field strength (about 9.81 m/s on Earth). The normal force here represents the force applied by the table against the object that prevents it from sinking through the table and requires that the table is sturdy enough to deliver this normal force without breaking. However, it is easy to assume that the normal force and weight are action-reaction force pairs (a common mistake). In this case, the normal force and weight need to be equal in magnitude to explain why there is no upward acceleration of the object. For example, a ball that bounces upwards accelerates upwards because the normal force acting on the ball is larger in magnitude than the weight of the ball. question: is the normal force equal to the force of gravity? answer: |
| Target Completion → | yes |

**Figure G.29:** Formatted dataset example for BoolQ

| | |
|---|---|
| Context → | The trend toward lower rents may seem surprising given that some communities in New York are bemoaning the loss of favorite local businesses to high rents. But, despite the recent softening, for many of these retailers there's still been too big a jump from the rental rates of the late 1970s, when their leases were signed. Certainly, the recent drop in prices doesn't mean Manhattan comes cheap. question: Manhattan comes cheap. true, false, or neither? answer: |
| Target Completion → | false |

**Figure G.30:** Formatted dataset example for CB

| | |
|---|---|
| Context → | The bet, which won him dinner for four, was regarding the existence and mass of the top quark, an elementary particle discovered in 1995. question: The Top Quark is the last of six flavors of quarks predicted by the standard model theory of particle physics. True or False? answer: |
| Target Completion → | False |

**Figure G.31:** Formatted dataset example for RTE

| | |
|---|---|
| Context → | An outfitter provided everything needed for the safari. Before his first walking holiday, he went to a specialist outfitter to buy some boots. question: Is the word 'outfitter' used in the same way in the two sentences above? answer: |
| Target Completion → | no |

**Figure G.32:** Formatted dataset example for WiC

```
            Context →   Final Exam with Answer Key
                        Instructions:  Please carefully read the following
                        passages.  For each passage, you must identify which noun
                        the pronoun marked in *bold* refers to.
                        =====
                        Passage:  Mr.  Moncrieff visited Chester's luxurious
                        New York apartment, thinking that it belonged to his son
                        Edward.  The result was that Mr.  Moncrieff has decided to
                        cancel Edward's allowance on the ground that he no longer
                        requires *his* financial support.
                        Question:  In the passage above, what does the pronoun
                        "*his*" refer to?
                        Answer:
```

| Target Completion → | mr.  moncrieff |
|---|---|

**Figure G.33:** Formatted dataset example for WSC

```
            Context →   Q: 'Nude Descending A Staircase' is perhaps the most famous
                        painting by which 20th century artist?

                        A:
```

| Target Completion → | MARCEL DUCHAMP |
|---|---|
| Target Completion → | r mutt |
| Target Completion → | duchamp |
| Target Completion → | marcel duchamp |
| Target Completion → | R.Mutt |
| Target Completion → | Marcel duChamp |
| Target Completion → | Henri-Robert-Marcel Duchamp |
| Target Completion → | Marcel du Champ |
| Target Completion → | henri robert marcel duchamp |
| Target Completion → | Duchampian |
| Target Completion → | Duchamp |
| Target Completion → | duchampian |
| Target Completion → | marcel du champ |
| Target Completion → | Marcel Duchamp |
| Target Completion → | MARCEL DUCHAMP |

**Figure G.34:** Formatted dataset example for TriviaQA. TriviaQA allows for multiple valid completions.

```
            Context →   Q: What school did burne hogarth establish?

                        A:
```

| Target Completion → | School of Visual Arts |
|---|---|

**Figure G.35:** Formatted dataset example for WebQA

```
            Context →   Keinesfalls dürfen diese für den kommerziellen Gebrauch
                        verwendet werden.  =
```

| Target Completion → | In no case may they be used for commercial purposes. |
|---|---|

**Figure G.36:** Formatted dataset example for De→En. This is the format for one- and few-shot learning, for this and other langauge tasks, the format for zero-shot learning is "Q: What is the {language} translation of {sentence} A: {translation}."

```
            Context →   In no case may they be used for commercial purposes.  =
```

| Target Completion → | Keinesfalls dürfen diese für den kommerziellen Gebrauch verwendet werden. |
|---|---|

**Figure G.37:** Formatted dataset example for En→De

| | | |
|---|---|---|
| Context → | Analysis of instar distributions of larval I. verticalis collected from a series of ponds also indicated that males were in more advanced instars than females. = | |
| Target Completion → | L'analyse de la distribution de fréquence des stades larvaires d'I. verticalis dans une série d'étangs a également démontré que les larves mâles étaient à des stades plus avancés que les larves femelles. | |

**Figure G.38:** Formatted dataset example for En→Fr

| | |
|---|---|
| Context → | L'analyse de la distribution de fréquence des stades larvaires d'I. verticalis dans une série d'étangs a également démontré que les larves mâles étaient à des stades plus avancés que les larves femelles. = |
| Target Completion → | Analysis of instar distributions of larval I. verticalis collected from a series of ponds also indicated that males were in more advanced instars than females. |

**Figure G.39:** Formatted dataset example for Fr→En

| | |
|---|---|
| Context → | The truth is that you want, at any price, and against the wishes of the peoples of Europe, to continue the negotiations for Turkey's accession to the European Union, despite Turkey's continuing refusal to recognise Cyprus and despite the fact that the democratic reforms are at a standstill. = |
| Target Completion → | Adevărul este că vă doriţi, cu orice preţ şi împotriva dorinţei europenilor, să continuaţi negocierile de aderare a Turciei la Uniunea Europeană, în ciuda refuzului continuu al Turciei de a recunoaşte Ciprul şi în ciuda faptului că reformele democratice au ajuns într-un punct mort. |

**Figure G.40:** Formatted dataset example for En→Ro

| | |
|---|---|
| Context → | Adevărul este că vă doriţi, cu orice preţ şi împotriva dorinţei europenilor, să continuaţi negocierile de aderare a Turciei la Uniunea Europeană, în ciuda refuzului continuu al Turciei de a recunoaşte Ciprul şi în ciuda faptului că reformele democratice au ajuns într-un punct mort. = |
| Target Completion → | The truth is that you want, at any price, and against the wishes of the peoples of Europe, to continue the negotiations for Turkey's accession to the European Union, despite Turkey's continuing refusal to recognise Cyprus and despite the fact that the democratic reforms are at a standstill. |

**Figure G.41:** Formatted dataset example for Ro→En

| | |
|---|---|
| Context → | Q: What is (2 * 4) * 6?<br>A: |
| Target Completion → | 48 |

**Figure G.42:** Formatted dataset example for Arithmetic 1DC

| | |
|---|---|
| Context → | Q: What is 17 minus 14?<br>A: |
| Target Completion → | 3 |

**Figure G.43:** Formatted dataset example for Arithmetic 2D-

| | |
|---|---|
| Context → | Q: What is 98 plus 45? |
| | A: |
| Target Completion → | 143 |

**Figure G.44:** Formatted dataset example for Arithmetic 2D+

| | |
|---|---|
| Context → | Q: What is 95 times 45? |
| | A: |
| Target Completion → | 4275 |

**Figure G.45:** Formatted dataset example for Arithmetic 2Dx

| | |
|---|---|
| Context → | Q: What is 509 minus 488? |
| | A: |
| Target Completion → | 21 |

**Figure G.46:** Formatted dataset example for Arithmetic 3D-

| | |
|---|---|
| Context → | Q: What is 556 plus 497? |
| | A: |
| Target Completion → | 1053 |

**Figure G.47:** Formatted dataset example for Arithmetic 3D+

| | |
|---|---|
| Context → | Q: What is 6209 minus 3365? |
| | A: |
| Target Completion → | 2844 |

**Figure G.48:** Formatted dataset example for Arithmetic 4D-

| | |
|---|---|
| Context → | Q: What is 9923 plus 617? |
| | A: |
| Target Completion → | 10540 |

**Figure G.49:** Formatted dataset example for Arithmetic 4D+

| | |
|---|---|
| Context → | Q: What is 40649 minus 78746? |
| | A: |
| Target Completion → | -38097 |

**Figure G.50:** Formatted dataset example for Arithmetic 5D−

| | |
|---|---|
| Context → | Q: What is 65360 plus 16204? |
| | A: |
| Target Completion → | 81564 |

**Figure G.51:** Formatted dataset example for Arithmetic 5D+

# H   Results on All Tasks for All Model Sizes

| Name | Metric | Split | Fine-tune SOTA | K | Zero-Shot | | | | | | | | One-Shot | | | | | | | | Few-Shot | | | | | | | | 175B (test server) |
|---|---|---|---|---|---|---|---|---|---|---|---|---|---|---|---|---|---|---|---|---|---|---|---|---|---|---|---|---|---|
| | | | | | Small | Med | Large | XL | 2.7B | 6.7B | 13B | 175B | Small | Med | Large | XL | 2.7B | 6.7B | 13B | 175B | Small | Med | Large | XL | 2.7B | 6.7B | 13B | 175B | |
| HellaSwag | acc | dev | 85.6 | 20 | 33.7 | 43.6 | 51.0 | 54.7 | 62.8 | 67.4 | 70.9 | 78.9 | 33.0 | 42.9 | 50.5 | 53.5 | 61.9 | 66.5 | 70.0 | 78.1 | 33.5 | 43.1 | 51.3 | 54.9 | 62.9 | 67.3 | 71.3 | 79.3 | |
| LAMBADA | acc | test | 68.0 | 15 | 42.7 | 54.3 | 60.4 | 63.6 | 67.1 | 70.3 | 72.5 | 76.2 | 22.0 | 47.1 | 52.6 | 58.3 | 61.1 | 65.4 | 69.0 | 72.5 | 22.0 | 40.4 | 63.2 | 57.0 | 78.1 | 79.1 | 81.3 | 86.4 | |
| LAMBADA | ppl | test | 8.63 | 15 | 18.6 | 9.09 | 6.53 | 5.44 | 4.60 | 4.00 | 3.56 | 3.00 | 165.0 | 11.6 | 8.29 | 6.46 | 5.53 | 4.61 | 4.06 | 3.35 | 165.0 | 27.6 | 6.63 | 7.45 | 2.89 | 2.56 | 2.56 | 1.92 | |
| StoryCloze | acc | test | 91.8 | 70 | 63.3 | 68.5 | 72.4 | 73.4 | 77.2 | 77.7 | 79.5 | 83.2 | 62.3 | 68.7 | 72.3 | 74.2 | 77.3 | 78.7 | 79.7 | 84.7 | 62.3 | 70.2 | 73.9 | 76.1 | 80.2 | 81.2 | 83.0 | 87.7 | |
| NQs | acc | test | 44.5 | 64 | 0.64 | 1.75 | 2.71 | 4.40 | 6.01 | 5.79 | 7.84 | 14.6 | 1.19 | 3.07 | 4.79 | 5.43 | 8.73 | 9.78 | 13.7 | 23.0 | 1.72 | 4.46 | 7.89 | 9.72 | 13.2 | 17.0 | 21.0 | 29.9 | |
| TriviaQA | acc | dev | 68.0 | 64 | 4.15 | 7.61 | 14.0 | 19.7 | 31.3 | 38.7 | 41.8 | 64.3 | 4.19 | 12.9 | 20.5 | 26.5 | 35.9 | 44.4 | 51.3 | 68.0 | 6.96 | 16.3 | 26.5 | 32.1 | 42.3 | 51.6 | 57.5 | 71.2 | 71.2 |
| WebQs | acc | test | 45.5 | 64 | 1.77 | 3.20 | 4.33 | 4.63 | 7.92 | 7.73 | 8.22 | 14.4 | 2.56 | 6.20 | 8.51 | 9.15 | 14.5 | 15.1 | 19.0 | 25.3 | 5.46 | 12.6 | 15.9 | 19.6 | 24.8 | 27.7 | 33.5 | 41.5 | |
| Ro→En 16 | BLEU-mb | test | 39.9 | 64 | 2.08 | 2.71 | 3.09 | 3.15 | 16.3 | 8.34 | 20.2 | 19.9 | 0.55 | 15.4 | 23.0 | 26.3 | 30.6 | 33.2 | 35.6 | 38.6 | 1.25 | 20.7 | 25.8 | 29.2 | 33.1 | 34.8 | 37.0 | 39.5 | |
| Ro→En 16 | BLEU-sb | test | | 64 | 2.39 | 3.08 | 3.49 | 3.56 | 16.8 | 8.75 | 20.8 | 20.9 | 0.65 | 15.9 | 23.6 | 26.8 | 31.3 | 34.2 | 36.7 | 40.0 | 1.40 | 21.3 | 26.6 | 30.1 | 34.3 | 36.2 | 38.4 | 41.3 | |
| En→Ro 16 | BLEU-mb | test | 38.5 | 64 | 2.14 | 2.65 | 2.53 | 2.50 | 3.46 | 4.24 | 5.32 | 14.1 | 0.35 | 3.30 | 7.89 | 8.72 | 13.2 | 15.1 | 17.3 | 20.6 | 1.25 | 5.90 | 9.33 | 10.7 | 14.3 | 16.3 | 18.0 | 21.0 | |
| En→Ro 16 | BLEU-sb | test | | 64 | 2.61 | 3.11 | 3.07 | 3.09 | 4.26 | 5.31 | 6.43 | 18.0 | 0.55 | 3.90 | 9.15 | 10.3 | 15.7 | 18.2 | 20.8 | 24.9 | 1.64 | 7.40 | 10.9 | 12.9 | 17.2 | 19.6 | 21.8 | 25.8 | |
| Fr→En 14 | BLEU-mb | test | 35.0 | 64 | 1.81 | 2.53 | 3.47 | 3.13 | 20.6 | 15.1 | 21.8 | 21.2 | 1.28 | 15.9 | 23.7 | 26.3 | 29.0 | 30.5 | 30.2 | 33.7 | 4.98 | 25.5 | 28.5 | 31.1 | 33.7 | 34.9 | 36.6 | 39.2 | |
| Fr→En 14 | BLEU-sb | test | | 64 | 2.29 | 2.99 | 3.90 | 3.60 | 21.2 | 15.5 | 22.4 | 21.9 | 1.50 | 16.3 | 24.4 | 27.0 | 30.0 | 31.6 | 31.4 | 35.6 | 5.30 | 26.2 | 29.5 | 32.2 | 35.1 | 36.4 | 38.3 | 41.4 | |
| En→Fr 14 | BLEU-mb | test | 45.6 | 64 | 1.74 | 2.16 | 2.73 | 2.15 | 15.1 | 8.82 | 12.0 | 25.2 | 0.49 | 8.00 | 14.8 | 15.9 | 20.3 | 23.3 | 24.9 | 28.3 | 4.08 | 14.5 | 19.3 | 21.5 | 24.9 | 27.3 | 29.5 | 32.6 | |
| En→Fr 14 | BLEU-sb | test | 45.9 | 64 | 2.44 | 2.75 | 3.54 | 2.82 | 19.3 | 11.4 | 15.3 | 31.3 | 0.81 | 10.0 | 18.2 | 19.3 | 24.7 | 28.3 | 30.1 | 34.1 | 5.31 | 18.0 | 23.6 | 26.1 | 30.3 | 33.3 | 35.5 | 39.9 | |
| De→En 16 | BLEU-mb | test | 40.2 | 64 | 2.06 | 2.87 | 3.41 | 3.63 | 21.5 | 17.3 | 23.0 | 27.2 | 0.83 | 16.2 | 22.5 | 24.7 | 28.2 | 30.7 | 33.0 | 30.4 | 3.25 | 22.7 | 26.2 | 29.2 | 32.7 | 34.8 | 37.3 | 40.6 | |
| De→En 16 | BLEU-sb | test | | 64 | 2.39 | 3.27 | 3.85 | 4.04 | 22.5 | 18.2 | 24.4 | 28.6 | 0.93 | 17.1 | 23.4 | 25.8 | 29.2 | 31.9 | 34.5 | 32.1 | 3.60 | 23.8 | 27.5 | 30.5 | 34.1 | 36.5 | 39.1 | 43.0 | |
| En→De 16 | BLEU-mb | test | 41.2 | 64 | 1.70 | 2.27 | 2.31 | 2.43 | 12.9 | 8.66 | 10.4 | 24.6 | 0.50 | 7.00 | 12.9 | 13.1 | 18.3 | 20.9 | 22.5 | 26.2 | 3.42 | 12.3 | 15.4 | 17.1 | 20.9 | 23.0 | 26.6 | 29.7 | |
| En→De 16 | BLEU-sb | test | 41.2 | 64 | 2.09 | 2.65 | 2.75 | 2.92 | 13.7 | 9.36 | 11.0 | 25.3 | 0.54 | 7.40 | 13.4 | 13.4 | 18.8 | 21.7 | 23.3 | 27.3 | 3.78 | 12.9 | 16.1 | 17.7 | 21.7 | 24.1 | 27.7 | 30.9 | |
| Winograd | acc | test | 93.8 | 7 | 66.3 | 72.9 | 74.7 | 76.9 | 82.4 | 85.7 | 87.9 | 88.3 | 63.4 | 68.5 | 72.9 | 76.9 | 82.4 | 84.6 | 86.1 | 89.7 | 63.4 | 67.4 | 73.6 | 76.9 | 84.3 | 85.4 | 82.4 | 88.6 | |
| Winogrande | acc | dev | 84.6 | 50 | 52.0 | 52.1 | 57.4 | 58.7 | 62.3 | 64.5 | 67.9 | 70.2 | 51.3 | 53.0 | 58.3 | 59.1 | 61.7 | 65.8 | 66.9 | 73.2 | 51.3 | 52.6 | 57.5 | 59.1 | 62.6 | 67.4 | 70.0 | 77.7 | |
| PIQA | acc | dev | 77.1 | 50 | 64.6 | 70.2 | 72.9 | 75.1 | 75.6 | 78.0 | 78.5 | 81.0 | 64.3 | 69.3 | 71.8 | 74.4 | 74.3 | 76.3 | 77.8 | 80.5 | 64.3 | 69.4 | 72.0 | 74.3 | 75.4 | 77.8 | 79.9 | 82.3 | 82.8 |
| ARC (Challenge) | acc | test | 78.5 | 50 | 26.6 | 29.5 | 31.8 | 35.5 | 38.0 | 41.4 | 43.7 | 51.4 | 25.5 | 30.2 | 31.6 | 36.4 | 38.4 | 41.5 | 43.1 | 53.2 | 25.5 | 28.4 | 32.3 | 36.7 | 39.5 | 43.7 | 44.8 | 51.5 | |
| ARC (Easy) | acc | test | 92.0 | 50 | 43.6 | 46.5 | 53.0 | 53.8 | 58.2 | 60.2 | 63.8 | 68.8 | 42.7 | 48.2 | 54.6 | 55.9 | 60.3 | 62.6 | 66.8 | 71.2 | 42.7 | 51.0 | 58.1 | 59.1 | 62.1 | 65.8 | 69.1 | 70.1 | |
| OpenBookQA | acc | test | 87.2 | 100 | 35.6 | 43.2 | 45.2 | 46.8 | 53.0 | 50.4 | 55.6 | 57.6 | 37.0 | 39.8 | 46.2 | 46.4 | 53.4 | 53.0 | 55.8 | 58.8 | 37.0 | 43.6 | 48.0 | 50.6 | 55.6 | 55.2 | 60.8 | 65.4 | |
| Quac | f1 | dev | 74.4 | 5 | 21.2 | 26.8 | 31.0 | 30.1 | 34.7 | 36.1 | 38.4 | 41.5 | 21.1 | 26.9 | 31.9 | 32.3 | 37.4 | 39.0 | 40.6 | 43.4 | 21.6 | 27.6 | 32.9 | 34.2 | 38.2 | 39.9 | 40.9 | 44.3 | |
| RACE-h | acc | test | 90.0 | 10 | 35.2 | 37.9 | 40.1 | 40.9 | 42.4 | 44.1 | 44.6 | 45.5 | 34.3 | 37.7 | 40.0 | 42.0 | 43.8 | 44.3 | 44.6 | 45.9 | 34.3 | 37.0 | 40.4 | 41.4 | 42.3 | 44.7 | 45.1 | 46.8 | |
| RACE-m | acc | test | 93.1 | 10 | 42.1 | 47.2 | 52.1 | 52.3 | 54.7 | 54.4 | 56.7 | 58.4 | 42.3 | 47.3 | 51.7 | 55.2 | 56.1 | 54.7 | 56.9 | 57.4 | 42.3 | 47.0 | 52.7 | 53.0 | 55.6 | 55.4 | 58.1 | 58.1 | |
| SQuADv2 | em | dev | 90.7 | 16 | 22.6 | 32.8 | 33.9 | 43.1 | 43.6 | 45.4 | 49.0 | 52.6 | 25.1 | 37.5 | 37.9 | 47.9 | 47.9 | 51.1 | 56.0 | 60.1 | 27.5 | 40.5 | 39.2 | 53.5 | 50.0 | 56.6 | 62.6 | 64.9 | |
| SQuADv2 | f1 | dev | 93.0 | 16 | 28.3 | 40.2 | 41.4 | 50.3 | 51.0 | 52.7 | 56.3 | 59.5 | 30.1 | 43.6 | 44.1 | 54.0 | 54.1 | 57.1 | 61.8 | 65.4 | 32.1 | 45.5 | 44.9 | 58.7 | 55.9 | 62.1 | 67.7 | 69.8 | |
| CoQA | f1 | dev | 90.7 | 5 | 34.5 | 55.0 | 61.8 | 65.3 | 71.1 | 72.8 | 76.3 | 81.5 | 30.6 | 52.1 | 61.6 | 66.1 | 71.8 | 75.1 | 77.9 | 84.0 | 31.1 | 52.0 | 62.7 | 66.8 | 73.2 | 77.3 | 79.9 | 85.0 | |
| DROP | f1 | dev | 89.1 | 20 | 9.40 | 13.6 | 14.4 | 16.4 | 19.7 | 17.0 | 24.0 | 23.6 | 11.7 | 18.1 | 20.9 | 23.0 | 26.4 | 27.3 | 29.2 | 34.3 | 12.9 | 18.7 | 24.0 | 25.6 | 29.7 | 29.7 | 32.3 | 36.5 | |
| BoolQ | acc | dev | 91.0 | 32 | 49.7 | 60.3 | 58.9 | 62.4 | 67.1 | 65.4 | 66.2 | 60.5 | 52.6 | 61.7 | 60.4 | 63.7 | 68.4 | 68.7 | 69.0 | 76.7 | 43.1 | 60.6 | 62.0 | 64.1 | 70.3 | 70.0 | 70.2 | 77.5 | 76.4 |
| CB | acc | dev | 96.9 | 32 | 0.00 | 32.1 | 8.93 | 19.6 | 19.6 | 28.6 | 19.6 | 46.4 | 55.4 | 53.6 | 53.6 | 48.2 | 57.1 | 33.9 | 55.4 | 64.3 | 42.9 | 58.9 | 53.6 | 69.6 | 67.9 | 60.7 | 66.1 | 82.1 | 75.6 |
| CB | f1 | dev | 93.9 | 32 | 0.00 | 29.3 | 11.4 | 17.4 | 22.4 | 25.1 | 20.3 | 42.8 | 60.1 | 39.8 | 45.6 | 37.5 | 45.7 | 28.5 | 44.6 | 52.5 | 26.1 | 40.4 | 32.6 | 48.3 | 45.7 | 44.6 | 46.0 | 57.2 | 52.0 |
| Copa | acc | dev | 94.8 | 32 | 66.0 | 68.0 | 73.0 | 77.0 | 76.0 | 80.0 | 84.0 | 91.0 | 62.0 | 64.0 | 66.0 | 74.0 | 76.0 | 82.0 | 86.0 | 87.0 | 67.0 | 64.0 | 72.0 | 77.0 | 83.0 | 83.0 | 86.0 | 92.0 | 92.0 |
| RTE | acc | dev | 92.5 | 32 | 47.7 | 49.8 | 48.4 | 56.0 | 46.6 | 55.2 | 62.8 | 63.5 | 53.1 | 47.3 | 49.5 | 49.5 | 54.9 | 54.9 | 56.3 | 70.4 | 52.3 | 48.4 | 46.9 | 50.9 | 56.3 | 49.5 | 60.6 | 72.9 | 69.0 |
| WiC | acc | dev | 76.1 | 32 | 0.00 | 0.00 | 0.00 | 0.00 | 0.00 | 0.00 | 0.00 | 0.00 | 50.0 | 50.3 | 50.3 | 49.2 | 49.4 | 50.3 | 50.0 | 48.6 | 49.8 | 55.0 | 53.0 | 53.0 | 51.6 | 53.1 | 51.1 | 55.3 | 49.4 |
| WSC | acc | dev | 93.8 | 32 | 59.6 | 56.7 | 65.4 | 61.5 | 66.3 | 60.6 | 64.4 | 65.4 | 58.7 | 58.7 | 60.6 | 62.5 | 66.3 | 66.3 | 66.3 | 69.2 | 58.7 | 60.6 | 54.8 | 49.0 | 62.5 | 67.3 | 75.0 | 75.0 | 80.1 |
| MultiRC | acc | dev | 62.3 | 32 | 4.72 | 9.65 | 12.3 | 13.6 | 14.3 | 18.4 | 24.2 | 27.6 | 4.72 | 9.65 | 12.3 | 13.6 | 14.3 | 18.4 | 24.2 | 27.6 | 6.09 | 11.8 | 16.8 | 20.8 | 24.7 | 23.8 | 25.0 | 32.5 | 30.5 |
| MultiRC | f1a | dev | 88.2 | 32 | 57.0 | 59.7 | 60.4 | 59.9 | 60.0 | 64.5 | 71.4 | 72.9 | 57.0 | 59.7 | 60.4 | 59.9 | 60.0 | 64.5 | 71.4 | 72.9 | 45.0 | 55.9 | 64.2 | 65.4 | 69.5 | 66.4 | 69.3 | 74.8 | 75.4 |
| ReCoRD | acc | dev | 92.5 | 32 | 70.8 | 78.5 | 82.1 | 84.1 | 86.2 | 88.6 | 89.0 | 90.2 | 69.8 | 77.0 | 80.7 | 83.0 | 85.9 | 88.0 | 88.8 | 90.2 | 69.8 | 77.2 | 81.3 | 83.1 | 86.6 | 87.9 | 88.9 | 89.0 | 90.2 |
| ReCoRD | f1 | dev | 93.3 | 32 | 71.9 | 79.2 | 82.8 | 85.2 | 87.3 | 89.5 | 90.4 | 91.0 | 70.7 | 77.8 | 81.6 | 83.9 | 86.8 | 88.8 | 89.7 | 91.2 | 70.7 | 77.9 | 82.1 | 84.0 | 87.5 | 88.8 | 89.8 | 90.1 | 91.1 |
| SuperGLUE | average | dev | 89.0 | 32 | 40.6 | 47.4 | 46.8 | 49.6 | 50.1 | 52.3 | 54.4 | 58.2 | 54.4 | 55.1 | 56.7 | 57.8 | 61.2 | 59.7 | 64.3 | 68.9 | 50.2 | 56.2 | 56.8 | 60.0 | 64.3 | 63.6 | 66.9 | 73.2 | 71.8 |
| ANLI R1 | acc | test | 73.8 | 50 | 33.4 | 34.2 | 33.4 | 33.4 | 34.2 | 32.3 | 33.2 | 34.6 | 32.1 | 31.6 | 31.9 | 34.6 | 30.6 | 31.6 | 32.7 | 32.0 | 32.1 | 32.5 | 30.9 | 32.5 | 33.5 | 33.1 | 33.3 | 36.8 | |
| ANLI R2 | acc | test | 50.7 | 50 | 33.2 | 31.9 | 33.3 | 33.3 | 33.8 | 33.5 | 33.5 | 35.4 | 35.7 | 33.7 | 33.2 | 32.7 | 32.7 | 33.9 | 33.9 | 33.9 | 35.7 | 33.8 | 32.1 | 31.4 | 32.6 | 33.3 | 32.6 | 34.0 | |
| ANLI R3 | acc | test | 48.3 | 50 | 33.6 | 34.0 | 33.8 | 33.4 | 35.3 | 34.8 | 34.4 | 34.5 | 35.0 | 32.6 | 33.0 | 33.9 | 34.1 | 33.1 | 32.5 | 35.1 | 35.0 | 34.4 | 35.1 | 36.0 | 32.7 | 33.9 | 34.5 | 40.2 | |
| 2D+ | acc | n/a | | 50 | 0.70 | 0.65 | 0.70 | 0.85 | 1.10 | 2.54 | 15.4 | 76.9 | 2.00 | 0.55 | 3.15 | 4.00 | 12.1 | 19.6 | 73.0 | 99.6 | 2.00 | 4.10 | 3.50 | 4.50 | 8.90 | 11.9 | 55.5 | 100.0 | |
| 2D- | acc | n/a | | 50 | 1.25 | 1.25 | 1.25 | 1.25 | 1.60 | 7.60 | 12.6 | 58.0 | 1.15 | 0.95 | 1.45 | 1.95 | 3.85 | 11.5 | 44.6 | 86.4 | 1.15 | 1.45 | 2.25 | 2.70 | 7.35 | 13.6 | 52.4 | 98.9 | |
| 3D+ | acc | n/a | | 50 | 0.10 | 0.10 | 0.05 | 0.10 | 0.10 | 0.25 | 1.40 | 34.2 | 0.15 | 0.00 | 0.10 | 0.30 | 0.45 | 0.95 | 15.4 | 65.5 | 0.15 | 0.45 | 0.30 | 0.55 | 0.75 | 0.90 | 8.40 | 80.4 | |
| 3D- | acc | n/a | | 50 | 0.05 | 0.05 | 0.05 | 0.05 | 0.05 | 0.45 | 1.35 | 48.3 | 0.05 | 0.15 | 0.25 | 0.30 | 0.55 | 1.60 | 6.15 | 78.7 | 0.05 | 0.10 | 0.15 | 0.35 | 0.65 | 1.05 | 9.20 | 94.2 | |
| 4D+ | acc | n/a | | 50 | 0.05 | 0.05 | 0.00 | 0.00 | 0.05 | 0.05 | 0.15 | 4.00 | 0.00 | 0.00 | 0.10 | 0.00 | 0.00 | 0.10 | 0.80 | 14.0 | 0.00 | 0.05 | 0.05 | 0.00 | 0.15 | 0.15 | 0.40 | 25.5 | |
| 4D- | acc | n/a | | 50 | 0.00 | 0.00 | 0.00 | 0.00 | 0.00 | 0.00 | 0.10 | 7.50 | 0.00 | 0.00 | 0.00 | 0.05 | 0.00 | 0.50 | 1.40 | 14.0 | 0.00 | 0.05 | 0.00 | 0.00 | 0.10 | 0.05 | 0.40 | 26.8 | |
| 5D+ | acc | n/a | | 50 | 0.00 | 0.00 | 0.00 | 0.00 | 0.00 | 0.00 | 0.00 | 0.65 | 0.00 | 0.00 | 0.00 | 0.00 | 0.00 | 0.00 | 0.05 | 3.45 | 0.00 | 0.00 | 0.00 | 0.00 | 0.00 | 0.00 | 0.05 | 9.30 | |
| 5D- | acc | n/a | | 50 | 0.00 | 0.00 | 0.00 | 0.00 | 0.00 | 0.00 | 0.00 | 0.80 | 0.00 | 0.00 | 0.00 | 0.00 | 0.00 | 0.00 | 0.05 | 3.75 | 0.00 | 0.00 | 0.00 | 0.00 | 0.00 | 0.00 | 0.00 | 9.90 | |
| 2Dx | acc | n/a | | 50 | 2.20 | 2.25 | 2.65 | 2.10 | 2.55 | 5.80 | 6.15 | 19.8 | 1.35 | 2.35 | 3.35 | 2.35 | 4.75 | 9.15 | 11.0 | 27.4 | 1.35 | 2.90 | 2.70 | 2.85 | 4.25 | 6.10 | 7.05 | 29.2 | |
| 1DC | acc | n/a | | 50 | 1.25 | 2.95 | 2.75 | 0.05 | 0.30 | 2.35 | 0.75 | 9.75 | 1.90 | 2.80 | 2.85 | 3.65 | 6.45 | 9.15 | 8.20 | 14.3 | 1.70 | 2.15 | 3.90 | 5.75 | 6.20 | 7.60 | 9.95 | 21.3 | |
| Cycled Letters | acc | n/a | | 100 | 0.62 | 0.71 | 2.85 | 0.00 | 0.63 | 1.35 | 2.58 | 3.66 | 1.67 | 4.36 | 5.68 | 6.46 | 6.25 | 9.41 | 15.1 | 21.7 | 4.63 | 9.27 | 10.7 | 14.5 | 16.7 | 21.9 | 27.7 | 37.9 | |
| Anagrams 1 | acc | n/a | | 100 | 0.10 | 0.14 | 0.40 | 0.00 | 0.27 | 0.69 | 1.16 | 2.28 | 0.21 | 0.61 | 1.12 | 1.27 | 1.60 | 2.72 | 3.72 | 8.62 | 0.50 | 1.27 | 2.13 | 3.05 | 3.81 | 5.49 | 8.38 | 15.1 | |
| Anagrams 2 | acc | n/a | | 100 | 0.81 | 1.21 | 2.69 | 0.01 | 1.71 | 3.75 | 4.53 | 8.91 | 1.19 | 2.62 | 4.70 | 4.77 | 6.97 | 10.2 | 14.6 | 25.9 | 1.94 | 4.80 | 7.59 | 9.87 | 12.6 | 18.9 | 25.6 | 39.7 | |
| Symbol Insertion | acc | n/a | | 100 | 0.00 | 0.00 | 0.10 | 0.00 | 0.45 | 0.42 | 0.89 | 8.26 | 0.03 | 0.05 | 0.57 | 1.18 | 1.67 | 3.46 | 6.62 | 45.4 | 0.11 | 0.28 | 2.19 | 4.18 | 6.61 | 11.0 | 27.3 | 67.2 | |
| Reversed Words | acc | n/a | | 100 | 0.00 | 0.01 | 0.01 | 0.01 | 0.02 | 0.03 | 0.03 | 0.09 | 0.02 | 0.01 | 0.01 | 0.00 | 0.05 | 0.07 | 0.11 | 0.48 | 0.00 | 0.05 | 0.00 | 0.17 | 0.24 | 0.30 | 0.42 | 0.44 | |
| SAT Analogies | acc | n/a | | 20 | 35.6 | 39.0 | 45.2 | 44.1 | 50.0 | 49.2 | 52.7 | 53.7 | 30.5 | 41.2 | 43.1 | 46.5 | 55.1 | 54.3 | 53.5 | 59.1 | 30.5 | 40.4 | 42.8 | 40.6 | 48.4 | 51.9 | 53.5 | 65.2 | |

**Table H.1:** Scores for every task, setting and model that we investigate in this paper.

**Figure H.1:** All results for all SuperGLUE tasks.

**Figure H.2:** Results for SAT task.

**Figure H.3:** All results for all Winograd tasks.

**Figure H.4:** All results for all Arithmetic tasks.

**Figure H.5:** All results for all Cloze and Completion tasks.

**Figure H.6:** All results for all Common Sense Reasoning tasks.

**Figure H.7:** All results for all QA tasks.

**Figure H.8:** All results for all Reading Comprehension tasks.

**Figure H.9:** All results for all ANLI rounds.

**Figure H.10:** All results for all Scramble tasks.

**Figure H.11:** All results for all Translation tasks.

## Footnotes

[1]https://spark.apache.org/docs/latest/api/python/pyspark.ml.html#pyspark.ml.feature.HashingTF