[Reviews · NeurIPS 2020]

Review 1

Summary and Contributions: The paper introduces GPT-3, a very large-scale Transformer language model of 175B parameters trained on 400B tokens from CommonCrawl data. The model obtains surprisingly effective results on zero-shot and few-shot scenario, without any finetuning. With only a prompt, or conditioning on a few examples, GPT-3 obtains strong performance on a wide variety of tasks, showing that large-scale language models, while only accessing isolated text data without any other modality and being trained on a very simple task, builds an impressive understanding of human natural language. I believe this work constitutes a seminal paper that significantly advances our understanding of what's possible in natural language understanding. It is one of the most interesting paper I have read in the deep learning era. I would fight for this paper to be accepted as an oral presentation at NeurIPS 2020 and I further recommend it for the best paper award.

Strengths: The paper in one of these research works that are simple conceptually (training a very large language model at scale) yet ground-breaking (redefines what we thought was possible). The amount of work behind this is enormous and the combination of simplicity, strong engineering work and new discovery makes it a very enjoyable paper to read. I have of course particularly enjoyed reading the part on the distinction of zero-/one-/few-shot learning and seeing the incredible capacity of the GPT-3 model. The fact that a very big neural net can perform a language task without any finetuning is definitely novel and in my opinion unforeseen. This takes us much closer to a system capable of performing multiple tasks at once with little to no supervision - as humans - and reveals a hint of what will be possible in the *near* future with large-scale self-supervised techniques, possibly combined with multiple modalities. Authors don't "just" train a large model and show its performance, they spend a lot of effort on trying to understand how it works by training multiple other language models of increasing size and showing the evolution of the performance; by providing an analysis of the performance wrt the number of examples in context among other things. The analysis part is overall very strong and interesting. The paper is very well written, with a lot of attention to details, and exemplifies good practice in paper writing by acknowledging for instance (i) a small "bug" in the data contamination analysis which is well explained and authors follow up on this by showing it does not impact that much performance (ii) that some of their unsupMT results might not be true SOTA (iii) as well as providing every detail necessary to reproduce the results without giving the impression to hide anything under the rug. This makes reading this paper a real pleasure. The analysis of the potential misuse of the GPT-3 model is comprehensive and particularly interesting. This was obviously an important part and the authors did a very good job at it. There are things not included in that review that are also great but it would become repetitive to mention that work was well done there: all the details given on data filtering, on model training in the Appendix, as well as the overlap statistics, the table on the compute used, the analysis of participant biases in the MTurk experiment on news generation, and finally the list of interesting samples etc. It is all very impressive.

Weaknesses: One could regret the lack of "finetuned" results, but I personally don't think it's necessary or even desirable as it is not the point of the paper (especially given the model is not bidirectional which could put it at a disadvantage). The goal of the paper is to focus on zero/one/few-shot learning, without fine-tuning (which means one model for everything, and very limited supervision). This is what is groundbreaking and I like the story this way. Finetuning results would imo only deviate the attention from the true novelty that is zero/one/few-shot learning.

Correctness: Yes the method is correct and well analyzed. Every detail is given to understand (if not reproduce) how experiments were run. The authors do not spend too much time on the implementation details while for sure this work requires an impressive amount of engineering effort. I think this is important to stress how much work this represents; behind the apparent simplicity (which is a great thing to have) there is a lot of strong engineering and research work that made this possible. The "Broader impacts" section and its analysis of the potential misuse of the model (biases, energy usage, news generation), could constitue a great NeurIPS paper in itself and is very well detailed. This section was particularly important for this paper and it is great to see the authors putting so much effort into the deep analysis of the model biases (race, gender, religion etc) as well as to see that GPT-3 can create short news-article that are (almost) indistinguishable from human-generated articles. Table C.2 is simply incredible work on analyzing overlap of all datasets used by the train set of the language model.

Clarity: The paper is very *very* well written. Explanations are simple and to the point, links to previous work are clear, the story is very fluent and builds naturally on previous work. The direction of research appears very clearly and this paper is a milestone in this very ambitious research plan.

Relation to Prior Work: Yes, the related work is very strong, and gives a lot of perspective on what has already been done in the literature, and how this paper contributes to the field.

Reproducibility: Yes

Additional Feedback: Congratulations to the authors for such an amazing study and paper. This is very exciting and constitutes a big step forward for the AI field and for NLP. POST AUTHOR RESPONSE: Thank you to the authors for answering our questions in the response. There is a clear consensus on the high quality and impact of this paper.


Review 2

Summary and Contributions: In this paper, the authors empirically demonstrate that increasing the model size -- in term of depth and width, and thus number of parameters -- of language models (LM) result in a better task-agnostic learner, which can zero/one/few-shots multiple well-know NLP tasks. - The authors use the same transformer base architecture as GPT-2, except for the Sparse attention (Child, et.al. 2019), which improve the model efficiency. They trained 8 models from 125 M to 175 B parameters to study the effect of the model size in the zero/one/few-shots settings. - The authors train the LM using 300 billion tokens from 5 sources (i.e., Common Crawl, WebText2, Book Collection 1 and 2, Wikipedia). - The authors evaluate the models' performance in a zero/one/few-shot setting on a large variety of NLP tasks such as LM perplexity, QA, CQA, SuperGLUE, MT, etc. Importantly, the zero/one/few-shots is done without fine-tuning the model, but by providing as context --priming-- the task-description (i.e., for zero-shot) or pairs of examples (one/few-shots), and making the model auto-regressively generate the response. As also clearly stated by the authors, this approach is not novel, since also GPT-2 used the same mechanism, but in this paper, the author extended the evaluation to way more tasks and showed that by increasing the model size the few-shot ability of the model greatly increases. - The authors compare the performance of the model to the current state-of-the-art and they highlight the advantages and disadvantages of the proposed model. I really appreciated the openness of the authors when they described their results, avoiding strong wording and objectively reporting when the model fails in the tasks. - The overall paper is well written and clear for a large audience and the broader impact of the paper is clear and well discussed. - The authors deeply analyse the potential misuse and ethical issue of the proposed model. I am personally not an expert in the bias/ethical ML, but in my perspective, the authors included a lot of insight and analysis of the model bias in different aspects (e.g. religion, race, gender).

Strengths: - the zero/one/few-shots methodology is impressive, although not novel per se (i.e. GPT-2), and it can have a big impact in many down-stream scenarios. - the evaluation includes a large variety of NLP tasks and SOTA baselines, and support the claim of the paper - the paper include an human evaluation over news article generation showing that human found hard (52% accuracy) to recognise which article is written by humans or the GPT-3 model (175B)

Weaknesses: - the authors already discussed most of the limitation of the current model (e.g. missing of bi-directional attection etc.). I found that one limitation could be the length of the context when increasing the number of shots. To elaborate, in some tasks (e.g. QA, summurization) where the input are entire articles, going beyond the 25/30 shots would be very challenging. GPT-3 already double (2048 tokens) the context size compare to GPT-2, but scaling very long inputs remains challenging, both in term of memory consumption and models inference (although the authors already use Sparse Transformer).

Correctness: Yes.

Clarity: Yes.

Relation to Prior Work: yes, to the best they could do in 8 pages. I think the citation format is not the NeurIPS 2020 template, but this can be easily change in the camera ready.

Reproducibility: Yes

Additional Feedback:


Review 3

Summary and Contributions: This paper pushes the boundaries of pre-trained autoregressive language models in terms of size of the model and size of the training data. Experiments show that the resulting model (GPT-3) is capable of attaining reasonable to SOTA performance on a variety of tasks simply by auto-completion (with or without being shown a few training examples, but in all cases without fine-tuning the actual weights).

Strengths: My takeaway from this work is that either 1. These large transformer models are able to achieve a surprisingly high level of reasoning just by "crunching data" 2. They aren't, and our testbeds should be updated to reflect that Either way, I think this work will spark yet another wave of advances in NLP, be it harder tasks/benchmarks or more efficient ways of achieving these results. Strong points of this work: - Extensive empirical evaluation on a large number of tasks - Comprehensive discussion of the limitations - Thorough broader impact statement

Weaknesses: Presentation: While I understand the difficulty of fitting this work in the fairly restrictive 8 pages Neurips format, I think the paper would benefit from including more of the ethics/impact discussion in the main body of the paper (and perhaps some of the experimental results could be moved to the appendix, as they are fairly predictable after the first few results tables). Reproducibility: This work is virtually irreproducible by all but a few major players in the research community (industry labs with large amounts of resources). Not necessarily a weakness per se, but a limitation worth mentioning nonetheless

Correctness: The only default in empirical methodology seems to be the "bug" mentioned in the appendix, which resulted in a partial leak of the test/validation data of some tasks into the training data. While the authors acknowledge this, and to some extent address this issue (by creating test sets clean of leaked examples), I think that this bug should be mentioned in the main text (as far as I can see, one has to read the appendix to find out)

Clarity: The paper is clearly written and understandable.

Relation to Prior Work: This work is well situated in relation with previous work (BERT, GPT-2, T-5)

Reproducibility: No

Additional Feedback: - The text in Figure 1.1 is too small to read, even on a screen (without zooming a lot)


Review 4

Summary and Contributions: The paper shows scaling up language models can achieve task-agnostic few-shot performances on various NLP tasks. Besides other promising results on various tasks and examples, this paper has a clear contribution to the community; industry-level, heavy engineering efforts, and their analyses on various aspects. I do appreciate such efforts and empirical findings described in the paper.

Strengths: The paper has the following strengths: (1) A comprehensive analysis has been made to evaluate the model in terms of task-agonistic behavior, memorization, and weakness based on the prior works/critiques made on the previous version of this work. (2) The empirical observations about the few-shot models’ capabilities are made (Figure 1.1 and Section 3.4), showing the scaling effects, limitations, and upper-bound of few-shot settings. (3) Experiments on different tasks in various applications indicate how much existing datasets/tasks are getting benefit from these huge-size language models and what kinds of operations (e.g., bidirectional, reasoning, external knowledge) should be done in the future toward that direction. (4) I pretty much enjoyed reading the Broader Impact section, which tries to adopt the feedback or criticism made from the earlier version of this work, including fairness and bias, energy usage, and potential misuse of the model. (5) For a perspective of usefulness, I think this can be applied to various NLP applications.

Weaknesses: Improvements from few shot LMs are not that surprising because it is mainly because the model uses more training data/parameters/computing resources. Since it is not fair to compare with relatively smaller models, the improvements themselves may not be the main contribution of this paper. In that manner, as an empirical paper with huge engineering efforts, this paper is worth to be accepted by NeurIPS, although more discussions about why still remain unexplored. For example, what types of linguistic capabilities could be achieved from this large language model for each task? Does better language modeling imply that it can do numerical reasoning (I saw some examples related to these but a few samples though), causal inferences, long-term coherence, stylistic variations, etc? Unfortunately, the paper seems to have huge media highlights by being seen as the general artificial intelligence or complete language understanding/generation system. I don’t believe a single NeurIPS paper can be like that but, then addressing more practical limitations rather than showing cherry picked ones or having more down ton in the writing seem to be necessary. There is a lack of scientific contributions and insights learned from this work. Compared to GPT2, the only difference made in this work is scaling up the training in terms of data size and comparing the model in three different settings: fine-tuning, few-shot, one-shot, and zero-shot. What scientific values does this paper bring to our community except for empirical observations found? In the experiment, does underperformance by GPT3 on NaturalQS shows GPT3 mean that it does not include any external knowledge in Wikipedia and their appropriate reasoning? Any more insights or error analysis? In the first figure of Figure 1.1., let’s say you use 175B x 1000 parameters, do you think the improvement from {zero,one,few}-shots still linearly increases? In the second figure of Figure 1.1., as the number of contexts increases, the degree of improvements from the few-shot GPT3 seems to be not that steep. Does this indicate the upper bound of the few-shot GPT3 is somewhere around K=100 or something? Also, please show me the zero/one-short cases as well.

Correctness: Please see many comments above.

Clarity: Yes, the paper is written well and easy to follow.

Relation to Prior Work: Yes.

Reproducibility: Yes

Additional Feedback:

[Author Response · NeurIPS 2020]

We thank the reviewers for their constructive and fair reviews. We do our best to address the points brought up by the
reviewers in the text below.

We agree that including the ethics and impact sections in the main body would have been preferable, given unlimited
space. Given the space constraints, we chose to discuss our bias analysis and other broader impacts work in the separate
Broader Impacts section, as we felt it was important to give this work a more thorough, contextual treatment, which
would have been difficult in the main body. We hope that future NeurIPS formats will be more supportive of a tighter
integration between the main body and Broader Impacts section.

One reviewer expressed curiosity about whether increasing scale will continue to yield results. As noted in the paper,
we do not see significant deviation from the power law trends across eight orders of magnitude. We expect an eventual
plateau, but we do not yet have evidence as to where this plateau will take place.

Finally, we appreciate the suggestions of adding additional studies of (1) fine-tuning, (2) the effect of context length, (3)
causal reasoning, (4) numerical reasoning, (5) stylistic variations and (6) long-term coherence. We agree that all of
these studies would be valuable, but felt that they could each merit their own papers to fully explore, so we leave them
as fertile avenues for future work.

[Meta-Review · NeurIPS 2020]

This work extends the Transformer language model architecture of GPT-2 by scaling it to > 170 billion parameters, resulting in a new model called GPT-3. The paper demonstrates that when provided with zero or few labeled examples to condition on, this large model is capable of performing a multitude of language tasks without any further changes to model parameters. While on most tasks the zero-shot/few-shot performance is behind SOTA, the novelty lies in the demonstrated strong zero/few shot performance on diverse tasks. Clarity of exposition is another strength of the paper. One limitation is lack of reproducibility due to the massive compute necessary to train the model. Another limitation is that the paper’s scientific insights are limited, and the contribution is largely engineering. However, the strong experimental findings, and the thorough analysis presented in the paper make it worthy of acceptance at NeurIPS. Regarding ethical concerns, a big language model can attract the interest of malicious actors. However, the paper has done a thorough job of addressing these concerns.